# Hypergravity Attenuates Reactivity in Primary Murine Astrocytes

**DOI:** 10.3390/biomedicines10081966

**Published:** 2022-08-13

**Authors:** Yannick Lichterfeld, Laura Kalinski, Sarah Schunk, Theresa Schmakeit, Sebastian Feles, Timo Frett, Harald Herrmann, Ruth Hemmersbach, Christian Liemersdorf

**Affiliations:** 1Department of Gravitational Biology, Institute of Aerospace Medicine, German Aerospace Center, 51147 Cologne, Germany; 2Department of Muscle and Bone Metabolism, Institute of Aerospace Medicine, German Aerospace Center, 51147 Cologne, Germany; 3Institute of Neuropathology, University of Erlangen, 91054 Erlangen, Germany

**Keywords:** neuroscience, primary astrocytes, astrocyte reactivity, astrogliosis, gravitational biology, neuronal regeneration, glial scarring, cytoskeletal remodeling, hypergravity

## Abstract

Neuronal activity is the key modulator of nearly every aspect of behavior, affecting cognition, learning, and memory as well as motion. Hence, disturbances of the transmission of synaptic signals are the main cause of many neurological disorders. Lesions to nervous tissues are associated with phenotypic changes mediated by astrocytes becoming reactive. Reactive astrocytes form the basis of astrogliosis and glial scar formation. Astrocyte reactivity is often targeted to inhibit axon dystrophy and thus promote neuronal regeneration. Here, we aim to understand the impact of gravitational loading induced by hypergravity to potentially modify key features of astrocyte reactivity. We exposed primary murine astrocytes as a model system closely resembling the in vivo reactivity phenotype on custom-built centrifuges for cultivation as well as for live-cell imaging under hypergravity conditions in a physiological range (2*g* and 10*g)*. We revealed spreading rates, migration velocities, and stellation to be diminished under 2*g* hypergravity. In contrast, proliferation and apoptosis rates were not affected. In particular, hypergravity attenuated reactivity induction. We observed cytoskeletal remodeling of actin filaments and microtubules under hypergravity. Hence, the reorganization of these key elements of cell structure demonstrates that fundamental mechanisms on shape and mobility of astrocytes are affected due to altered gravity conditions. In future experiments, potential target molecules for pharmacological interventions that attenuate astrocytic reactivity will be investigated. The ultimate goal is to enhance neuronal regeneration for novel therapeutic approaches.

## 1. Introduction

Astrocytes are the most abundant glial cell type in the adult brain depicting a distinctive morphology (“star-cells”). They exhibit versatile functions including ion and neurotransmitter homeostasis, trophic support, structural integrity of the tissue, neuronal transmission or synaptic plasticity regulation, and formation of the blood brain barrier (BBB) [1,2,3]. Astrocytes in the central nervous system (CNS) respond to injury or illnesses of all severities with changes on a molecular, cellular, and functional level. Normal, naive astrocytes exhibit several important functions to maintain CNS tissue homeostasis, such as trophic and structural support, extracellular clearance, vasoconstrictive regulation, secretion of growth factors, or the fine-tuning of neuronal plasticity alterations [2,4,5]. Variations in the astrocyte phenotype are indeed observed between different neurodegenerative diseases such as Alzheimer’s disease, Huntington’s disease, and Multiple Sclerosis [3,6]. Reactive astrocytes have beneficial functions during acute phases, following neuronal injury including containment of inflammatory cells and potential infection, degradation of apoptotic cells, and the inhibition of widespread tissue regeneration of damaged areas. However, during progressive astrogliosis and finally persistent glial scarring they show detrimental effects in effectively inhibiting neuronal regeneration [7].

One of the highly investigated but still unresolved topics in modern neuroscience is the regeneration of neuronal projections beyond the injury or lesion site since astrocytes shift toward a reactive phenotype (reactive astrogliosis), which triggers the formation of glial scar tissue in proximity to the impairment locus [8,9,10,11]. A lesion in the nervous tissues, for instance by traumatic brain injuries (TBI), spinal cord injury (SCI), stroke, epileptic seizures, neurodegenerative diseases or cancerous tissue evagination is most often associated with persistent glial scarring, neuronal death, and severe impairments of the patient. Neuronal projections connecting the areas surrounding the lesion site get damaged, degenerate, and retract therefore becoming inactive. The regeneration of neuronal projections beyond the glial scar is extremely ineffective, resulting in sustained damage to the tissue and subsequently in impairments in cognitive and/or motor functions [12,13,14,15]. Potential therapeutic applications to counteract these restrictions are therefore of utmost importance.

Depending on the mammalian species, 33–66% of the brain consists of glia cells [16]. So far, various types of glial cells have been identified including astrocytes, oligodendrocytes, microglia, and NG2 cells. The variety of functions of glial cells still remain elusive, nonetheless glial cells are indispensable for undisturbed and controlled neuronal function. Astrocytic reactivity is a highly coordinated response to changes in the CNS, where intercellular responses within the tissue are essential [9]. Upon becoming reactive, astrocytes show an altered gene expression pattern, which induces a shift in their phenotypic appearance as well as cellular behavior [5,6,17]. Reactive astrocytes obtain features that are not pursued in naive, non-reactive astrocytes, such as changes in gene expression, hyper-proliferative states, apoptosis reduction, enhanced migratory behavior, altered cytokine secretion, multinucleated cells with irregularly shaped nuclei and hypertrophy [7,18,19,20,21]. A common marker to detect astrocytic reactivity is glial fibrillary acidic protein (GFAP), an intermediate filament protein that is already expressed in normal, dormant astrocytes. Upon induction of a reactive phenotype, however, GFAP is significantly upregulated [3,22,23,24]. Besides reactive astrocytes, the glial scar consists of meningeal fibroblasts, oligodendrocyte precursors, and microglia [10]. The glial-scar inherent cells generate a chemical barrier by secretion of growth-inhibiting molecules. These effects cause major and long-lasting or persistent restrictions, such as the loss of neuronal signaling or even permanent paralysis [11].

A novel approach to counteract induction of astrocyte reactivity investigated here is the impact of increased gravitational loading, usually referred to as hypergravity, on primary murine astrocytes. Changes in gravitational loading can serve as a mediator to induce mechanical stress. Astrocytes are known to be able to sense and react to mechanical cues or stresses and respond by mechanosensitve channel activation and intracellular Ca-waves [25,26,27]. Different gravity paradigms can in turn alter cytoskeletal remodeling of glial cells to partially inhibit reactive astrogliosis. Hypergravity exposure in the physiological relevant range of 2*g*–10*g* can be used as a mild, temporary, and spatially defined approach to alter the neuronal cytoskeleton. Earth’s gravitational acceleration is approximated with 9.81 m/s^2^ and designated by the unit *g*. Research in altered gravity is important to identify the impact of gravity on signaling processes of biosystems, gravity-sensing mechanisms, and gravity-mediated orientation of organisms in their environment [28,29]. Furthermore, Earth’s gravity is the only environmental stimulus that has remained constant throughout the entire evolution [29].

The extreme environmental conditions of space, especially microgravity (i.e., weightlessness), exert diverse effects on the human body. Astronauts and model animals have exhibited morphological changes of the CNS as well as functional aberrancies and impairments on the cellular level [30,31]. It has been reported that space flight has a negative impact on cognitive performance and mental focus [32]. Moreover, a decrease in neuronal plasticity and shifts in grey matter distribution were detected with MRI scans of astronaut brains [33,34]. These detrimental effects on human neuronal health make it inevitable to find measures to counteract the effects of microgravity. One attractive option is to employ hypergravity as a potential countermeasure during or following exposure to microgravity [35].

To better understand and investigate whether hypergravity exposure can potentially inhibit reactive astrogliosis, we used primary murine cortical astrocytes as a model. These cells are closely related to human astrocytes in vivo in neuronal tissues, and they can be cultured for several months. Primary astrocytes are able to become reactive, secrete cytokines, exhibit calcium-waves, and depict similar proliferation and apoptosis rates as compared to their in vivo counterparts [36]. Thus, these cells serve as an ideal model system for in vitro astrogliosis. Here, we show that hypergravity did not induce a reactive phenotype, but it was able to inhibit certain aspects of astrocyte reactivity.

## 2. Materials and Methods

### 2.1. Primary Murine Cortical Astrocyte Isolation & Mouse Lines

Primary murine astrocytes were generated from the cortices of 18.5 day old C57Bl/6J wildtype or LifeAct-GFP heterozygous (LAGFP/+) mouse embryos. The culture was performed according to Craig and Banker 1988. In brief, gravid mice were sacrificed and the embryonic brains were dissected. Under a stereomicroscope the meninges and the hippocampus were removed from the cortices that were transferred to centrifuge tubes (Corning, Germany). The cells were trypsinized with a solution of 0.05% Trypsin/HBSS (PAN Biotech, Aidenbach, Germany) for 15 min at 37 °C and subsequently washed 3 times with warm HBSS/HEPES (Merck, Darmstadt, Germany). The cells were dissociated by pipetting through glass Pasteur pipets (Roth, Karlsruhe, Germany), followed by dissociation through fire-polished Pasteur pipets. The single cell suspension was seeded into T75 flasks (ThermoFisher, Waltham, MA, USA) and grown in MEM medium (PAN Biotech, Aidenbach, Germany), containing 10% FCS at 37 °C, 5% CO_2_, and >90% humidity in an incubator. After 24 h, the medium was exchanged and a fresh medium was supplied every 7 days. The cells were cultured until reaching 80–90% confluency before splitting them in a 1:3 ratio. All experiments were performed with primary astrocytes in passage 2 (P2).

### 2.2. Exposure to Hypergravity Conditions

#### 2.2.1. MuSIC Incubator Centrifuge

The DLR Multi Sample Incubator Centrifuge (MuSIC) is a custom-built swing-out centrifuge inside a cell culture incubator (Binder C260, Tuttlingen, Germany) (Appendix A). Incubation at controlled environmental conditions (temperature, CO_2_, humidity) can be applied for a variety of cell culture vessels that can be mounted onto specialized gondolas designed to swing out upon centrifugation to ensure a perpendicular direction of the resulting gravity vector on the samples and to prevent unwanted effects by shear forces. The centrifuge can be operated at up to 50*g* radial acceleration with a high accuracy of ±0.1*g*. The acceleration ramp-up and down time can be adjusted, reaching 2*g* in approximately 10 s via a brushless DC motor and a variable frequency drive. The samples can be exposed ranging from seconds to weeks. Control samples can be placed stationary on the bottom of the incubator, where they will be maintained for control but at 1*g* of Earth’s gravity.

Cells were exposed to 2*g* or 10*g* hypergravity or 1*g* as a control. Long-term spreading rates were measured by seeding astrocytes onto PLL-coated glass cover slips with 1.5 × 10^4^ cells/cover slip and exposing them to 2*g* hypergravity. The centrifuge was stopped daily, to fix the samples in 4% PFA for 20 min at 37 °C. For the long-term wound-healing assay, the centrifuge was stopped daily for a duration of approximately 30 min for each sample during which all remaining samples were still subjected to hypergravity. The cells were transported in a transport incubator at 37 °C and 5% CO_2_ for image acquisition (Zeiss, Jena, Germany 20× air NA 0.4 objective). After imaging, the samples were put back into the centrifuge to continue the hypergravity exposure.

#### 2.2.2. The Hyperscope Live-Cell Imaging Platform on the Human Centrifuge

The analyses of dynamic processes were performed on the Hyperscope live-cell imaging platform (Zeiss Axio Observer.Z1 epifluorescence microscope, Jena, Germany) on the DLR human centrifuge (SAHC, DLR: envihab, Cologne, Germany) (Appendix A). The microscope is a fully automated fluorescent system equipped with cell incubation units for optimal temperature (37 °C), CO_2_ (5%) and relative humidity (>90%) control, and a hardware focus stabilization system. Image acquisition can be controlled remotely during centrifuge operations. The setup is mounted on a swing-out platform to verify the perpendicular direction of the resulting gravity vector onto the cells. Appropriate LED excitation wavelength and corresponding filters sets were used (LEDs 385 nm, 475 nm, 555 nm, 590 nm, 630 nm). Images were captured with a high-sensitivity CMOS camera suitable for fast (128 fps) time-lapse acquisitions (Zeiss AxioCam 702, Jena, Germany). The centrifuge is a short-arm human centrifuge with 3.80 m radius and a maximum acceleration for the Hyperscope facility of 4*g*. The ramp-up or down time for the centrifuge to reach 2*g* or to stop again was approximately 2 min.

To achieve the best comparability, two paired live-cell imaging sessions for 2*g* hypergravity and 1*g* as a control were performed in close temporal proximity. For each biological replicate, two flasks of primary astrocytes at passage 1 derived from the same mouse were used to be split for 1*g* and 2*g* samples, respectively.

##### Live Spreading Rates

Initial spreading rates were measured by seeding astrocytes onto 8-well slides (ibidi, Gräfelfing, Germany) with 6 × 10^3^ cells/well. Images were acquired immediately after cell seeding with increments of 30 min for 5 h with a 25× objective (Zeiss, Jena, Germany 25× Oil NA 0.8) using DIC contrast. The hypergravity exposure started in the morning, and the 1*g* control was imaged with identical settings in the afternoon of the same day.

##### Wound-Healing Assay

To estimate astrocyte migration rates, a wound-healing assay (“scratch-assay”) was performed. The cells were seeded onto 3.5 cm diameter dishes with a 4-well silicone insert (ibidi, Gräfelfing, Germany). Insert stages were 500 µm wide and 4 mm long, thus 4 defined cell-free areas were uncovered upon removal of the cell culture insert with minimal harm to surrounding cells. For each biological replicate we thus had four identical cell-free areas that were imaged in the same session. Images were acquired daily with a 20× objective (Zeiss NA 0.4 LD, Ph2, Jena, Germany) at phase contrast for long-term migration analysis. Using the classical scratch assay in which the cell-free area is scratched by hand, cells at the scratch border are damaged, thus inducing astrocyte reactivity. To circumvent this, for the following experiments an ibidi cell culture chamber was used (ibidi, Gräfelfing, Germany). A silicone scaffold was sealed onto a 3.5 cm petri dish (ibidi, Gräfelfing, Germany) and primary murine astrocytes were grown to confluency in each well. For each biological replicate we had 4 identical cell-free areas that could be imaged in the same session. For the live-cell migration analyses, tiled images were acquired with a 20× air objective with increments of 30 min over a duration of 22 h or 36 h using the microscope settings as described before (see Section 2.2.2 and its subsections). The migration speed of astrocytes in the wound-healing assays was calculated from the linear regression lines. The slopes of the regression lines could be seen as the scratch closing speed with the unit %/h. The cells migrated from two sides in the 500 µm wide cell-free area. For this reason, cell migration velocities in this assay could be calculated as $Speed^{Migration}=\frac{Slope\times 5}{2}$.

##### LifeAct-GFP Dynamics

The dynamics of F-actin rearrangements were visualized in LifeAct-GFP expressing astrocytes derived from heterozygous LifeAct-GFP transgenic mice during exposure to 2*g* hypergravity. The cells were cultured on 8-well slides (ibidi, Gräfelfing, Germany) and images were acquired with increments of 2.5 min as previously described (see Section 2.2.2 and its subsections). GFP was excited at 475 nm with 2% LED intensity to minimize phototoxicity, and fluorescence emission was detected via specific emission filters for GFP with a fast CMOS camera (see Section 2.2.2).

### 2.3. Staining & Image Analysis

#### 2.3.1. Immunofluorescence Microscopy

Cells fixed on coverslips were washed with Tris-buffered saline (TBS) and permeabilized with TBS-T (TBS + 0.1% Tween20) for 1 min. Autofluorescence was quenched with 50 mM NH_4_Cl for 10 min at RT and coverslips were washed. Unspecific binding sites were blocked using TBS-T + 10% normal goat serum at 4 °C overnight. Primary antibodies were diluted in TBS-T + 10% normal goat serum to antibody-specific concentrations and incubated overnight at 4 °C. On the following day, the cells were washed three times in cold TBS-T and afterwards incubated for 2 h at RT with secondary antibodies diluted in TBS-T + 10% normal goat serum at 1:2000. After washing, DNA was stained with DAPI and the coverslips were mounted onto microscope slides using Everbrite mounting medium (Biotium, Fremont, CA, USA).

#### 2.3.2. Proliferation and Apoptosis Assays

Astrocytic cells were seeded on coverslips in 24-well plates and exposed to 2*g* hypergravity in the MuSIC incubator centrifuge for 6, 12, 24, 48, 72, 96, and 120 h, respectively. For each time point, 1*g* controls were placed on the bottom of the MuSIC incubator. For evaluation of apoptosis levels, live cells were stained using Annexin-V-ATTO488 (1:20, Abcam, Cambridge, UK), followed by propidium iodide (1:10, ThermoFisher, Waltham, MA, USA) incubation. Cells were then washed and fixed. For evaluation of proliferation levels, cells were stained with an anti-Ki67 antibody (1:500, Thermo Fisher, Waltham, MA, USA) and an anti-rabbit secondary antibody coupled to an ATTO488 dye (ThermoFisher, Waltham, MA, USA) as well as Phalloidin-ATTO633 (1:250, Attotec, Siegen, Germany). Images were acquired with a 25× Oil objective (NA 0.8) on an Axio Observer.Z1 (Zeiss, Jena, Germany) epifluorescence microscope. Images of whole coverslips were acquired by tiled images that were stitched together by the Zeiss Zen software. A negative control was used to verify signal specificity omitting the primary antibody, using only the secondary antibody coupled to the respective fluorescent dye. Microscope settings adjusted for the negative control was used for all subsequent samples that were identically stained. Positive cells were manually counted, using Zeiss Zen software (Zeiss, Jena, Germany).

#### 2.3.3. Cytoskeletal Components and Reactivity Markers

For the identification of cytoskeletal components and reactivity markers, the cells were immediately fixed and equal numbers of coverslips were stained using specific antibodies. For the assessments of reactivity marker intensity levels, anti-GFAP (1:500, Synaptic Systems, Göttingen, Germany) and anti-LZK (MAP3K13; 1:500, Sigma-Aldrich, Darmstadt, Germany) as well as anti-vimentin (1:1000, Sigma-Aldrich, Darmstadt, Germany) and anti-nestin (1:100, Merck Millipore, Darmstadt, Germany) were used, respectively. As secondary antibodies anti-guinea pig or anti-chicken ATTO488 and anti-rabbit or anti-mouse ATTO550 (1:1000, Attotec, Siegen, Germany) were used. All samples were additionally incubated with Phalloidin-ATTO633 (1:250, Attotec, Siegen, Germany). Images were acquired as described before in Section 2.3.2. The mean intensities as grey scale values were measured for each traced cell with the cell boundary set as a new region of interest (ROI).

For the assessment of cytoskeletal components via STimulated Emission Depletion (STED) microscopy, Phalloidin-ATTO542 conjugates (1:250, Attotec, Siegen, Germany) were used in combination with an anti-α-tubulin primary antibody (1:500, Sigma, Darmstadt, Germany) and an anti-mouse secondary antibody coupled to an Abberior STAR RED dye (Abberior, Göttingen, Germany). Images were acquired with an Infinity Line STED microscope (Abberior, Göttingen, Germany) with pulsed laser excitation at 485 nm and 561 nm and spectral detectors (APDs, Abberior, Göttingen, Germany). The STED laser at 774 nm and 50% laser power was used to specifically deplete photon emission, which lead to approximate resolutions of 100 nm for microtubules and 250 nm for actin filaments.

#### 2.3.4. Image Analysis

Images were acquired for multiple purposes as described above (see Section 2.2.2, Section 2.3, Section 2.3.1, Section 2.3.2 and Section 2.3.3). The images were analyzed using the Zeiss Zen image analysis software (Zeiss, Jena, Germany). We developed semi-automated analysis routines especially for the quantification of spreading rates and migration velocities under hypergravity. For cellular morphology analysis, the Zeiss Zen Image Analysis Wizard was employed. An object class was created in the Phalloidin-stained fluorescence channel and it was set to be segmented by variance-based thresholding. The minimum variance was manually adjusted for every experiment to detect the Phalloidin-stained cell perimeter and the maximum variance was set to the highest value. Through this algorithm and using the “fill holes” function, most stained cells could be detected reliably and with detailed cell perimeter resolution. A region filter with a minimum object area of 200 µm^2^ was employed to filter out debris and staining artifacts. As a last step, the image and all detected objects were manually controlled for artifacts and multicellular clusters, which were removed to attain only single cells for the morphological analysis. Spreading rates were calculated by cell area increase over time in µm^2^, while migration rates were measured by tracing the cell-free areas of the wound-healing assays to calculate the cellular migration speed in µm/h.

##### Morphological Parameters

To assess the morphological features of cells and nuclei respectively, several parameters were measured via Zeiss Zen 3 image analysis software (Zeiss, Jena, Germany). The area was measured as a region excluding any holes in µm^2^. The diameter in µm was calculated by the formula sq(4/pi × area). The circularity was estimated by the formula sq(4 × area). To measure the Feret minima and maxima two straight lines were positioned on opposite sides of the object similar to a sliding caliper at 32 angle positions. The corresponding distance was measured for each angle position with the highest values defined as the Feret maximum and vice versa, with the lowest values for the Feret minimum. The ratio is given by dividing minimum/maximum.

##### Statistical Analysis

Analyzed cells were derived from three individual astrocyte cultures isolated from three gravid mice. The embryos ranged in number from only 1 to 11 for each gravid mouse. All embryos from the same mother were pooled into one culture, thus we received a representative distribution of cells and putative astrocyte subtypes.

Statistical analysis was performed in GraphPad Prism 9 (GraphPad Software, San Diego, CA, USA). The data was collected, a test for outliers (ROUT method, 1%) was conducted, and identified outliers were removed. A paired *t*-test was performed for all paired data (cell area, morphology parameters, western blot intensities, proliferation rate, apoptosis rate, Phalloidin signal intensities), while a nonparametric Mann–Whitney U test was used to determine statistical significance in independent samples (spreading rate, migration velocity, reactivity marker signal intensities). Mean values are depicted with SEM indicated by error bars. Measured changes were considered to be significant with a value of *p* < 0.05.

### 2.4. Biochemical Assessment of Protein Levels

For biochemical analyses, cells were grown in 6-well plates (ThermoFisher, Waltham, MA, USA) and exposed to hypergravity in the MuSIC centrifuge. The cells were washed with ice-cold PBS, and Triton lysis buffer (TLB: 150 mM NaCl, 50 mM Tris-HCl, Triton X-100, pH 7,4) was added to each well. The cells were scraped from the wells (Corning, NY, USA), transferred into a 1.5 mL centrifuge tube (Eppendorf, Hamburg, Germany), and centrifuged for 15 min at 14,000× *g* and 4 °C. The protein concentration was measured by the BCA protein assay, according to the manufacturer’s instructions (ThermoFisher, Waltham, MA, USA). Cell pellets were taken up in 1x SDS-loading buffer.

The lysates were subjected to discontinuous SDS-PAGE with 10 µg of protein per lane on 10% acrylamide-gels (BioRad, München, Germany) and blotted onto activated PVDF membranes (Merck, Darmstadt, Germany) in a tank blot system overnight at 4 °C and 20 V in Towbin buffer (BioRad, München, Germany). The gels were stained with Coomassie Brilliant Blue (Sigma, Darmstadt, Germany) and the membranes were stained with Ponceau S (Sigma, Darmstadt, Germany). The membranes were washed, unspecific binding was blocked by 5% non-fat milk power in NCP (Sigma, Darmstadt, Germany), and primary antibodies were diluted at antibody-specific concentrations in blocking buffer. Anti-GFAP (1:500, Synaptic Systems, Göttingen, Germany), anti-vimentin (1:1000, Sigma, Darmstadt, Germany), and anti-FAK (1:500, Cell Signaling, Danvers, MA, USA) antibodies were used to detect specific proteins of interest. The membranes were incubated with the primary antibody solutions at 4 °C overnight, followed by washing with NCP and incubation of species–specific secondary antibodies conjugated to horseradish peroxidase (Jackson, WY, USA) at a concentration of 1:2000 in blocking buffer for 2 h at RT. Specific signals were detected by enhanced chemiluminescence (BioRad, München, Germany) with a digital detector (Intas, Göttingen, Germany). To ensure comparability, 1*g* and 2*g* samples were processed in parallel on the same membrane. Band intensities were measured using ImageStudio software (LI-COR, Lincoln, NE, USA) and specific signals were normalized using signals of the housekeeping gene GAPDH. T-tests were performed in Excel (Microsoft, Redmond, WA, USA) to determine statistical significance. Results were regarded significant with *p* < 0.05. Values were shown as SEM.

## 3. Results

Primary astrocytes were cultured from embryonic murine cortices to yield high purity cultures that closely resemble astrocytes in situ in neuronal tissues and thus serve as an ideal in vitro model system for reactive astrogliosis. Astrogliosis in vivo occurs in line with induction of reactivity, a phenotypic change with characteristic features, such as hypertrophy, increased cell migration, hyperproliferation, increased cellular maintenance (decreased apoptosis), gene expression profile changes, chromatin remodeling, and cytokine secretion.

We exposed primary astrocytes at an early passage (P2) to altered gravity on our custom-designed experimental platforms. Performing experiments under altered gravity conditions requires meticulous controls, as gravity is a relatively weak environmental factor that needs to be investigated with vigilance, controlling for other environmental stimuli that otherwise might disturb the measurements. Our custom-built devices to expose cells to defined gravity conditions were designed to precisely control gravity levels (accuracy 0.1*g*) without generating vibrations, temperature fluctuations or osmotic changes due to media evaporation during rotation. The cells were grown in well-plates filled with medium to avoid cell shear by fluid movements and sealed with fluid-tight but gas-permeable membranes. The well-plates were mounted on gondolas inside the incubator-centrifuge that were allowed to swing out freely, thus ensuring a uniform perpendicular direction of the resulting gravity force on the cells. The controls were placed inside the bottom of the same incubator, with exactly the same environmental conditions as the cells exposed to hypergravity.

We aimed to analyze the impact of increased gravitational forces, i.e., hypergravity, on the previously mentioned processes and its potential to be used as a novel approach to inhibit the shift from dormant to reactive astrocytes that is coinciding with astrogliosis. Deciphering the underlying mechanisms of these alterations might lead to the identification of target genes or pathways, e.g., for novel pharmacological interventions to eventually treat patients suffering from neurological disorders or injuries.

### 3.1. Increased Gravitational Load Induces Spreading Deficits in Primary Astrocytes

Primary astrocytes were identified by staining for glial fibrillary acidic protein (GFAP) (Figure 1). In addition, GFAP levels function as a measure for astrocyte reactivity, as described above. F-actin was used to evaluate general cell morphology as well as for cytoskeletal rearrangements necessary for cell adhesion or migration.

In general, astrocytes were able to adhere, grow, and proliferate under hypergravity conditions without obvious impairments compared to the 1*g* controls.

Cell spreading is, similar to most alterations in cell shape, dependent on the tightly coordinated action of different cytoskeletal components. Alterations in the spreading process can thus be associated with alterations in cytoskeletal remodeling that is necessary to promote the structural support for morphological rearrangements of any given cell.

Primary astrocytes exposed to increased gravitational load in the physiological range of 2*g* demonstrated a decrease in cell area compared to 1*g* normal Earth’s gravity controls (Figure 2). The cells showed spreading deficits during exposure to 2*g* in an incubator centrifuge (Appendix A) with consistent area reduction of approximately 21% at 24 h and 18% over the course of 48 h (Figure 2A,B). A similar area-increase from 1d to 2d of 46.3% (±6.8%) at 1*g* compared to 51.5% (±5.2%) at 2*g* could be observed in culture. Thus, spreading rates were diminished, but spreading velocities over 24 h were not influenced.

This observation led us to the question if initial spreading rates were also affected or if an adaptation phase was required. Live-cell imaging immediately following cell attachment under the influence of 2*g* hypergravity employing the Hyperscope live-cell imaging microscope platform installed on a swing-out stage on the DLR large human centrifuge (Appendix A) revealed a decrease in initial spreading of approximately 45% after 5 h (3517 µm^2^ at 1*g* vs. 1576 µm^2^ at 2*g*) (Figure 2C,D). Astrocyte mean area increased by 355% at 1*g*, while cells exposed to 2*g* enlarged their area by 211% during this initial time immediately following cell attachment to the culture slide. Thus, astrocytes reacted gravity-sensitive and were unable to spread to a similar diameter as under normal gravity conditions. This deficit in cell area enlargement had no significant lag phase and persisted for at least 96 h under continuous hypergravity.

After becoming reactive, astrocytes undergo morphological changes, such as the enlargement of the cell body [37]. A decrease in cell area is contrary to the cell enlargement phenotypes observed in reactive astrogliosis.

### 3.2. Acute and Persistently Impaired Migration Speed of Astrocytes Exposed to 2g Hypergravity

Cell spreading is a morphological feature that requires dynamic cytoskeletal rearrangements, which are further mandatory for the initiation of cell migration. Decreasing cell spreading capabilities could furthermore lead to diminished cell migration. For astrocytes, such a behavior would indicate a reduced potential for astrogliosis induction, where increased astrocytic migration toward the lesion site is a well-conserved feature.

Indeed, we observed markedly reduced migrational velocities in a wound-healing assay with astrocytes exposed to 2*g* hypergravity in an incubator-centrifuge (Figure 3 and Appendix A). During the course of 5 days, images were acquired every 24 h in a 2D wound-healing assay (“scratch assay”). Astrocyte migration toward the cell-free area was significantly inhibited by approximately 15% after 5 days (Figure 3). The curves diverged more pronounced during the first 3 days, however, cells migrated at their respective steady pace afterwards.

To ameliorate the factor of the short potential adaptation time during image acquisition, and to focus on the initial fast acting dynamics in astrocyte migration behavior, a wound-healing assay was performed on the live-cell imaging Hyperscope platform on the DLR large human centrifuge (Figure 4 and Appendix A). Acute changes in cell migration were confirmed with a high impact of approximately 35% inhibition of migrational velocity during the first 22 h imaged with 30 min increments (Figure 4A,B). Following a 2.5 h “lag phase” in cells adapting to both 1*g* and 2*g*, migration was disturbed as indicated by dispersing linear regressions. During the lag phase migration velocities were similar in cells exposed to 1*g* or 2*g* conditions with 3.6 µm/h at 1*g* and 3.0 µm/h at 2*g*. Following the lag phase, the cells slowed down considerably and migrated under normal 1*g* gravity conditions with a velocity of 1.3 µm/h compared to 0.8 µm/h for the 2*g* sample (Figure 4B).

To investigate the influence of acute changes in gravitational loading conditions and the re-adaptation durations, we set up an intermittent wound-healing assay, in which the migrational velocity was measured at normal 1*g* conditions for 12 h before rapidly switching to 2*g* for 12 h (with an acceleration ramp-up time of approximately 2 min), followed by a re-adaptation phase of 12 h under normal 1*g* conditions (Figure 4C–F). This 1*g*-2*g*-1*g* intermittent gravity paradigm revealed a clear dependence and adaptation of migrational speed on the gravitational load. The cells reduced their migrational speed considerably from approximately 1.25 µm/h to 0.43 µm/h during hypergravity exposure (Figure 4C–F). The change in migration velocity was not instantaneous, but repeating adaptive “lag phases” were identified (Figure 4E,F). As indicated by the migration velocities that stayed constant before adapting to the new gravity loading condition, distinct lag periods were determined with approximately 1 h adaptation time from 1*g* to 2*g* hypergravity (Figure 4E) and 2 h re-adaption period from 2*g* hypergravity back to 1*g* normal gravity (Figure 4F). This change in gravitational force showed a stronger response for the switch from 1g to 2*g*, i.e., approximately 76% reduction in migration speed (from 1.25 µm/h at 1*g* to 0.43 µm/h at 2*g*) compared to the reverse switch from 2*g* to 1*g* as the speed exhibited during hypergravity was significantly sustained at 0.43 µm/h before returning to baseline levels of 1.28 µm/h upon re-adaptation to 1*g*.

### 3.3. Morphology Alterations upon Hypergravity Exposure Revealed Decreased Cell Polarity

Cell shape and directional migration is significantly inhibited under 2*g* hypergravity (compare Figure 2, Figure 3 and Figure 4). To gain a better understanding of astrocytic architecture and their polarization following the increase in gravitational load, key morphological aspects were analyzed in detail using Zeiss Zen image analysis software. During migration or induction of a reactive phenotype, astrocytes are known to become hypertrophic with elongated and irregularly shaped somata. Later on, they transform into scar-forming astrocytes and develop a stellate morphology [37].

To assess changes in cell morphology, e.g., due to processes in reactive astrogliosis, such as polarization, hypertrophy, or initiation of migration, the minimal and maximal distances of the opposing cell boundaries in a straight line are measured by the Feret minima and maxima (Figure 5). Circularity of the cultured astrocytes under normal and increased gravity conditions was calculated as a scope of regularity as well (Figure 5).

Astrocytes under hypergravity in the physiological range of 2*g* depicted decreased Feret minima as well as maxima (Figure 5A,B). The cells thus decreased in size but did not elongate or polarize. Cells becoming increasingly dissimilar would otherwise depict Feret ratios approaching zero. Rather, astrocytes increased in their circularity during the first day of exposure to hypergravity, which did reverse, however, on the second day (Figure 5C). Otherwise, cells that are hypertrophic, polarized, or migrating have a comparably lower circularity. For migration initiation, lamellipodial protrusions have to be formed. As a consequence, more elongated, seemingly irregular cell shapes were obtained. What distinguishes the circularity values measured in cells from the Feret distances measured in the same sample is that the circularity also considers cells that have a highly serrated boundary because of filopodia and other types of protrusions. In particular, hypertrophic cells exhibit similar Feret minima and maxima and additionally a lower circularity compared to cells with a smoother cell boundary. Thus, within the first 24 h, the morphology of astrocytes exposed to hypergravity changed toward a less polar appearance, resulting from decreased Feret minima and maxima and increased circularity (Figure 5C). During the course of the following 24 h, a switch in circularity was revealed under the influence of 2*g* hypergravity, as the cells became more irregular. Nonetheless, Feret distance values remained decreased. This might indicate that more small protrusions such as filopodia or membrane ruffles had formed at the cell boundaries.

In summary, astrocytes did alter their morphology in line with the inhibition of cell spreading and migration to a less polarized and less stellate cell shape. Changes observed are small but highly significant as large numbers of cells were evaluated. Hence, the described morphological features can be used to evaluate changes in cytoskeletal rearrangements due to altered gravitational loads.

### 3.4. Actin Filament and Microtubule Dynamics Are Affected by 2g Hypergravity

In the current study so far, we investigated parameters that are highly dependent on cytoskeletal rearrangements and dynamic adaptation processes. Especially cell migration requires the formation of a leading edge with adhesion to the substrate and simultaneous retraction and de-adhesion at the trailing edge. The actin-myosin system hereby provides the protrusive force necessary for cell movement. In cell culture, astrocytic protruding processes exhibit a slim outer rim of filamentous actin with occasional ruffles. In comparison to other extending or migrating cells, there are differences in the actin organization of polarizing and hypertrophic astrocytes after injury. Astrocytes respond by a drastic upregulation of the actin regulators α-actinin and its ligand paladin, which are known to reorganize actin filaments through crosslinking and bundling [37].

Investigations on dynamic changes in the F-actin network can be visualized live with little interference on actin filament stability and polymerization by employing primary astrocytes isolated from transgenic LifeAct-GFP expressing mouse embryos (mouse line LifeAct-GFP (Tg(CAG-GFP)#Rows)) [38,39,40].

We have visualized fluorescently labeled actin filaments live in cells during the switch from 1*g* to 2*g* hypergravity employing the Hyperscope microscope on the DLR human centrifuge (Figure 6 and Appendix A). Fluorescence excitation intensities and exposure times for frame rates of 2.5 min over a duration of 3 h were carefully tested and adjusted for signal stability and photobleaching. Under these conditions, we did not observe any detrimental effects on cell viability.

The F-actin rich structures visualized by LifeAct-GFP included lamellipodial and filopodial protrusions, stress fibers, and membrane ruffles. Notably, all these structures did rearrange quickly in response to altered gravity. When comparing cells in the same field-of-view (FoV) under static 1*g* conditions and again during hypergravity exposure, acute changes were observed. A considerable number of lamellipodia retracted in response to 2*g* hypergravity over the course of 30 min with filopodial retraction fibers remaining in place (Figure 6A–C). This observation is in line with the decrease in spreading rates and cell area described above. The lamellipodial retraction velocity on the other hand was faster and advanced nearly instantaneously without an obvious lag phase in contrast to the spreading or migrational changes (see Figure 2 and Figure 4).

F-actin rich membrane ruffles were observed less frequently under the influence of hypergravity (Figure 6B and Appendix A). Nonetheless, focal adhesions also formed under hypergravity conditions, however, focal adhesions were increasingly concentrated at the cell center (Figure 6B,C) [41].

Stress fibers appeared unimpaired with no hints for changes in filament number, altered lengths, diameter or orientation between individual filaments (Figure 6D,E). The increase in circularity described before could be verified in LifeAct-GFP expressing cells that formed smoother cell borders and retracted longer protrusions (Figure 6E).

In addition, F-actin levels were quantified by measurement of the mean fluorescence intensity of Phalloidin-labeled F-actin structures of astrocytes exposed to normal 1*g* or 2*g* hypergravity over 3 days on the MuSIC incubator centrifuge (Appendix A). Following exposure, the cells were fixed, and actin was labeled with fluorescently conjugated Phalloidin. The outline of individual actin filaments was traced and the mean fluorescence intensities were measured for each cell.

F-actin structures in cells normally grown at 1*g* Earth’s gravity enlarge and polarize over the time course of 2 days corresponding to increasing F-actin levels, but decrease after 3 days upon reaching their final spreading dimensions. Astrocytes exposed to 2*g* hypergravity showed, however, significantly increased F-actin intensities after 24 h of exposure in line with high turnover rates. Nevertheless, F-actin levels remained elevated over consecutive days under the influence of 2*g* (Appendix A).

Actin dynamics appeared to be modified in zones of high rearrangements, such as membranous (lamellipodia, filopodia, ruffles) and sub-membranous structures (focal adhesions), while stress fibers as force-bearing cytoskeletal elements were left undisturbed (Figure 6). Further, F-actin levels (Appendix A) were higher under hypergravity and remained elevated in cells exposed to 2*g* hypergravity.

While other growing or migrating cells such as fibroblasts and neurons possess a prominent actin meshwork and fibers in their periphery, most actin filaments in scar-forming astrocytes are instead concentrated around the cell body [37].

Live visualization of actin filaments revealed an influence of altered gravity conditions on the glial cell cytoskeleton. Stable structures, such as stress fibers remained undisturbed, while dynamic structures were changing due to increased gravitational load. Due to the fact that F-actin structures were affected by altered gravitational loads, microtubules were expected to be affected as well. Changes in shape, location, quantity, and distribution of cytoskeletal elements related to the tubulin- and actin-cytoskeleton are notoriously difficult to quantify. We performed super-resolution microscopy employing the STED (STimulated Emission Depletion) methodology to be able to detect fine architectural changes in the actin filament and microtubule network of astrocytes that have been exposed to normal 1*g* or 2*g* as well as 10*g* hypergravity for 24 h in the MuSIC incubator centrifuge. Due to super-resolution techniques, we could identify fine architectural changes in astrocytes exposed to gravitational loads in the physiologically relevant range of 2*g*. We wanted to further investigate the gravity-dependency for these intricate changes in filament networks and have thus performed an additional excessive exposure of the cells to 10-fold the normal gravity level (10*g*).

We achieved a resolution for microtubules of approximately 100 nm and 250 nm for actin filaments in xy-dimensions. Astrocytes under normal 1*g* conditions showed multiple radially distributed protrusive elements with the well-known basis of a dense cortical actin network extending stress fibers structurally supported by a less dense network of microtubules alongside the basis of each protrusion (Figure 7A–D). On the other hand, astrocytes under hypergravity responded by reorganizing the cytoskeletal networks (Figure 7E–H). Especially actin filaments were prominently visualized in cortical cell regions. Central or perinuclear actin stress fibers were observed less frequently under hypergravity conditions. Microtubules became denser but were still well-structured under hypergravity conditions with their density depending on the gravitational load, as this effect was more pronounced under 10*g* than under 2*g* conditions. As can be seen in the magnified inserts, microtubules were present at lamellipodial protrusions to the edge of the cells under hypergravity rather than on the basis of the protrusion as under 1*g* control conditions (Figure 7D,H,L). Larger, force-bearing elements, such as actin stress fibers or microtubules in the cells center remained unchanged, as only dynamic areas with high turnover showed changes in response to hypergravity.

These findings indicate that due to hypergravity exposure actin filaments retracted from the cell perimeter, while microtubules exhibited increased polymerization for enhanced structural support. Dynamic rearrangements of the astrocytic cytoskeletal elements were elevated. The finding that lamellipodial protrusions were mostly retracted supports the observations made during live-cell imaging of transgenic LifeAct-GFP expressing astrocytes (Figure 6).

### 3.5. Stable Expression Levels of Major Cytoskeletal Components

The effects described above were based on changes in cytoskeletal elements. Especially highly dynamic systems, such as actin filaments, were impacted. However, also focal adhesions or intermediate filaments (IFs) might play a decisive role in the measured changes in cell polarity, spreading, and migration rates documented above. Thus, we performed western blot analysis of cell lysates from primary astrocytes exposed to 1*g* or 2*g* hypergravity for 1, 2, and 3 days on the MuSIC incubator-centrifuge (Appendix A).

Intermediate filaments are not only important for structural integrity of cells but also for transducing mechanical cues into cellular responses [42]. In reactive astrogliosis, such a role of intermediate filaments in signal transduction has been emphasized. Notably, GFAP was identified as the key marker for astrocytic reactivity alongside other IFs, such as vimentin. Therefore, we investigated the expression levels of GFAP and vimentin on the protein level to gain insight into structural rearrangements as well as astrocyte reactivity. Expression of GFAP and vimentin showed no significant changes over the course of 3 days under 2*g* hypergravity compared to 1*g* normal gravity (Figure 8). The stable expression of GFAP and vimentin is in line with the results obtained in this study (compare Figure 1).

The levels of FAK expression as a marker protein for focal adhesion structures remained stable under increased gravitational loading even over three days (Figure 8). Thus, focal adhesions seemed stable also under increased gravity, which reflects previous results from live-cell and super-resolution microscopy (compare Figure 6 and Figure 7).

### 3.6. Proliferation and Apoptosis Rates Are Not Affected by Hypergravity

Assessment of a complex multi-state phenotype, such as astrocyte reactivity, requires consideration of as many relevant parameters as possible. As stated above, among the changes that occur in astrocytic behavior upon shifting towards a reactive state, well-known reactive phenotypes include hyperproliferation and increase in cellular maintenance, i.e., inhibition of apoptotic cell death. Furthermore, altered gravity, especially microgravity in spaceflight applications has been shown to induce detrimental changes in cell survival in various cell types, including fibroblasts, lymphocytes, or several glioma cell lines [43,44,45,46]. These important features of cellular viability need to be considered to identify potential features of astrocytic reactivity induction. We thus assessed both processes by immunofluorescent signal intensity measurements of key markers of apoptosis, necrosis, and proliferation.

The image analysis revealed that hypergravity does not influence cell viability or proliferative potential in astrocytes (Figure 9). Separating viable cells from early and late apoptotic cells was achieved by immunolabeling Annexin V, a protein selectively interacting with membrane-associated phosphatidylserine, which is only present on the extracellular membrane upon disturbances or ruptures in the membrane as they occur during apoptosis induction [47]. Thus, cells labeled with Annexin V conjugated to the fluorescent dye Atto488 already induced apoptosis but did not progress to late apoptotic stages, i.e., necrosis. During necrosis the nuclear membrane is ruptured as well, which will enable propidium iodide (PI) to intercalate into DNA fragments. Astrocytes are cells well-known for their homeostatic potential, which reflects in their low apoptosis rate during their native state (Figure 9A–C). Very low percentages of apoptotic (approximately 3%) or necrotic (approximately 0.5–1%) cells were measured for both normal 1*g* as well as 2*g* increased gravity conditions. Astrocytic survival was therefore not impacted by hypergravity in the physiological range of 2*g*.

Hyperproliferation, especially in proximity to the lesion site, is a key feature of reactive astrogliosis [48]. On a more technical note, the inhibition of migration velocity described above (compare Figure 3 and Figure 4) could in principle be caused by attenuated proliferation, as the closure of the cell-free area requires imaging for several days. To control for the possibility that the proliferative potential of astrocytes might influence our measurement of migration speed, we assessed proliferative states using Ki67, a well-established marker protein exclusively expressed in nuclei of replicating cells [49]. Astrocytes typically exhibit very slow growth rates, which could be observed in our measurements as well (Figure 9D,E). Indeed, hypergravity did not induce changes in the proliferative potential of astrocytes: Under both 1*g* and 2*g* hypergravity we observed proliferation rates of approximately 4% (Figure 9D,E). Neither hyperproliferation nor increased maintenance or inhibition of apoptosis were observed. These findings agree with the results described above indicating that hypergravity does not induce a reactive phenotype in astrocytes. Furthermore, the results obtained in the wound-healing assay were exclusively relying on cell migration processes, as proliferation rates were very similar on both 1*g* and 2*g* conditions.

### 3.7. Hypergravity Attenuates Astrocyte Reactivity

An often neglected feature of cells is the remodeling not only of their cell shape but also of their nuclear morphology. The cell nucleus is a dynamic organelle connected to F-actin and intermediate filaments, which might respond to changes in gravitational loading (compare Figure 2, Figure 6 and Figure 7). Nuclear reorganization is necessary, for instance to unfold chromatin to induce gene expression. In astrocyte irregularities of nuclear morphology or multinucleated cells are a sign of the shift from the usual dormant and homeostatic state toward a reactive state that requires gene expression changes [37,48]. We assessed the nuclear morphology, following exposure to 2*g* hypergravity or 1*g* as controls on the MuSIC incubator-centrifuge according to parameters that would demonstrate enlargements, irregularities, or lobed nuclei. In addition, nuclei per cell were counted using high-resolution imaging of DAPI. Image analysis was conducted via Zeiss Zen image software with semi-automated nuclear tracing, counting, and morphological assessment.

Nuclear morphology was largely unchanged and only few multinucleated cells were counted for either 1*g* or 2*g* conditions (Figure 10). Nuclear plasticity was observed between 1 and 2 days of exposure with a diminished area of approximately 29% at 1*g* and 26% at 2*g*, and a diminished diameter of approximately 15% at 1*g* and 13% at 2*g*. Thus, nuclei of astrocytes did not enlarge after exposure to 1 or 2 days of continuous hypergravity in the physiological range of 2*g* (Figure 10A,B). Nuclei also did not become irregular as can be acknowledged by their circularity that decreased from 1 to 2 days approximately 5% at 1*g* and 2% at 2g (Figure 10C). The nuclear morphology, however, changed in its Feret minimum, i.e., the shortest distance between two points within the nucleus, that decreased over the course of 24 h with 20% at 1*g* and 12% at 2*g*. The Feret maximum decreased only slightly between the days of exposure with 9% at 1*g* and 12% at 2*g*. Accordingly, the Feret ratio between minimum and maximum changed, which indicates an increased elongated nuclear shape possibly due to the cells struggle to start migrating but lacking the necessary cytoskeletal stability as described above (Figure 10D–F). The number of nuclei per cell is a measure for cell division accuracy, mutations and also for a reactive state, in case of astrocytes. The number of cells with more than one nucleus was, however, similar in both 1*g* and 2*g* gravity loads (Figure 10G).

Nuclear morphology and number per cell served as indicators of astrocyte reactivity. Hypergravity failed to induce irregular or lobed nuclei or multinucleated cells that could be mediated by altered gene expression in the reactive state.

One of the hallmarks of glial scar formation is a change from dormant, homeostatic to reactive astrocyte states. Astrocytic reactivity features a multitude of phenotypes described above. One strong indicator for astrocytes shifting from the usual phenotype to a reactive state is the change in cell morphology leading to hypertrophy and increased cell migration and driven by gene expression changes including the upregulation of the intermediate filaments GFAP, vimentin and nestin.

GFAP is the prevalent marker employed to identify astrocytes in general as well as to detect increased levels in astrocytic reactivity [3,22,23,24]. Several neurological diseases are associated with reactive astrogliosis and GFAP overexpression, often correlating with the severity of the impairment [50]. Other regulated genes include vimentin, nestin, leucine zipper kinase (LZK), β-catenin, n-cadherin, paladin, and matrix metalloproteinase-13 (MMP13) [5,7,48].

Vimentin shows different expression patterns during pathological conditions and is upregulated in the case of reactivity [7]. The expression of nestin is induced during neurological pathologies and has been shown to be upregulated by astrocytes in their reactive state [7]. Leucine zipper kinase (LZK or MAP3K13) is an enzyme, which is upregulated in the nucleus after a CNS injury [7,48]. The deletion of MAP3K13 in astrocytes of adult mice reduces astrogliosis and impairs scar formation, whereas overexpression enhances astrogliosis and scar formation [51]. Hence, the aforementioned marker proteins were highly suitable for further investigation of the reactive state of astrocytes exposed to altered gravity conditions.

In the current study, we chose several marker proteins (GFAP, vimentin, nestin, LZK) to ensure a reliable identification of the complex reactive state of the astrocytes in culture. We exposed primary astrocytes to 2*g* hypergravity and 1*g* normal gravity as a control for a total of 5 days in the MuSIC incubator-centrifuge (Appendix A) before fixation, immunofluorescence staining, and high-resolution epifluorescence imaging. The number of GFAP-high expressing, i.e., reactive cells was counted and the mean fluorescence intensity values for all reactivity markers were estimated.

Reactive astrocytes were identified by augmented expression of the investigated marker proteins. Especially upregulation of GFAP was very prominent and reactive cells were clearly distinguished from native cells (Figure 11). After six hours of 2*g* hypergravity, ~7.6% of cells showed an upregulated GFAP signal in contrast to ~9.3% of the control (18% reduction in 2*g* compared to 1*g*). In both gravity conditions the GFAP expression increased steadily with a peak being reached at 48 h. At this time point ~12.0% of cells in the 1*g* controls were reactive compared to ~11.2% under hypergravity. Over the next three days the GFAP expression in the 1*g* control as well as the hypergravity exposed cells decreased again, reaching ~9.3% in the control and ~8.5% in the hypergravity sample at the end of the experiment. At any time point astrocytes exposed to hypergravity depicted a lower percentage of GFAP high-expressing cells than the 1*g* control samples (Figure 11A,B).

Quantification of mean fluorescence intensities for the marker proteins GFAP, vimentin, nestin, and LZK revealed similar results as for the numbers of cells expressing high levels of GFAP (Figure 11C–F). The mean fluorescence intensity for GFAP increased nearly linearly from 6 h to 48 h where it reached a peak in both the control and the hypergravity samples (Figure 11C). In the 1*g* sample, the 48 h intensity value was 706% of the starting intensity measured at the 6 h time point. In comparison, the hypergravity exposed cells only increased their GFAP fluorescence intensity by 417% after 48 h. Over the following 3 days both the 1*g* control and the 2*g* sample showed a reduction in GFAP fluorescence intensity, reaching approximately baseline levels after 120 h. The divergence between the two conditions increased steadily up to the 48 h time point and decreased steadily from 48 h to 120 h.

LZK fluorescence intensity showed an increase from 6 h to a maximum at 48 h, analogous to the values for GFAP intensities (Figure 11D). The control samples increased fluorescence intensities by 152% from 6 h to 48 h, whereas the hypergravity exposed cells had a 109% increase over the same time. The LZK fluorescence decreased in both samples from 48 h to 96 h, reaching 49% increase for the 1*g* sample and 60% increase for the 2*g* sample as compared to the 6 h starting values. From 96 h to 120 h both samples reached approximately steady levels.

The vimentin fluorescence intensity of the 2*g* hypergravity sample exhibited an on average lower fluorescence intensity compared to the 1*g* sample. The cells at 2g depicted an already 58% lower starting value after 6 h compared to the 1*g* controls (Figure 11E). Under 2*g* hypergravity conditions, the vimentin fluorescence intensity values peaked after 48 h of exposure, with an 81% increase compared to the start. After 72 h of hypergravity exposure, vimentin fluorescence reached an intensity comparable to that at the start of the experiment, with a minor decrease from 72 h to 120 h. The control samples, in comparison, exhibited an increase in fluorescence from 6 h to 24 h by 32%, temporarily reached baseline levels at 48 h, before raising to maximum levels of 60% increase after 72 h. From 72 h to 120 h a steady decrease was measured, reaching a 26% reduction at the last measured time point as compared to the start at 6 h.

The nestin fluorescence intensity values were comparable to vimentin levels (Figure 11F). Fluorescence intensities under hypergravity showed a lower starting value compared to the controls, starting at 6 h with a 41% lower fluorescence intensity. The nestin fluorescence intensity peaked after 48 h in the hypergravity sample with a 37% increase. From 72 h to 120 h a slight decline in the fluorescence values was observed, ending with an 18% reduction compared to the starting value. In the 1*g* control, the fluorescence first decreased by 19% from 6 h to 12 h before reaching a maximum of 12% increased fluorescence after 72 h. From this time point on, the nestin fluorescence values decreased steadily, reaching a minimum of 26% reduction compared to the starting point at 6 h, which was the only point in the control samples lower than the maximum in the hypergravity samples.

Astrocytes in culture were able to become reactive and depicted clear reactivity marker expression compared to minor levels in dormant and homeostatic cells (Figure 11). The investigated reactivity markers all exhibited the same trend of lower GFAP-positive cell counts or mean fluorescence intensity levels for all marker proteins under 2*g* hypergravity. Therefore, hypergravity did not induce a shift to astrocyte reactivity, but rather slightly diminished the expression of genes involved in reactive astrogliosis.

In summary, astrocytes responded to hypergravity in the physiologically relevant range of 2*g* by remaining in their homeostatic state or even reducing their reactivity state. This phenomenon is potentially helpful to identify target genes and pathways, which could be, for instance, pharmacologically stimulated in order to inhibit reactive astrogliosis in vitro as well as in vivo. The ultimate goal would be to inhibit glial scar formation and thus enhance neuronal regeneration.

## 4. Discussion

Research on neuronal regeneration following nervous tissue injuries has become ever increasing with a rising number of patients and still very limited therapy options [12]. Patients of spinal cord injuries or head trauma often have to suffer through long-lasting or even permanent disabilities with complete loss-of-function. The function and inhibitory properties of the glial scar have been under debate for decades. The focus of recent studies has shifted from neuronal outgrowth toward the role of astrocytes in reactive astrogliosis and further in glial scar formation that majorly inhibits the regenerative potential. First its role as a sole dystrophic structure for neuronal regeneration was postulated [52,53] before the dual function of glial scar cells was revealed with both beneficial features, especially during acute phases of nervous tissue injury and detrimental features mostly during long-term and chronic phases of neuronal lesions [15,54].

We have only just begun to understand the intricate roles of reactive astrocytes in neuronal regeneration. Complete ablation of the glial scar has failed to enhance the progression of neuronal regeneration [55,56,57]. The ineffective repair after injury is a consequence of both cell intrinsic properties as well as extracellular components inhibiting outgrowth [58]. Thus, the ultimate aim is to partially inhibit astrocyte reactivity and glial scarring that will allow for inflammatory suppression near the lesion core during acute phases but also to diminish the inhibitory environment during chronic phases of neuronal injury progression. Ideally, the new therapeutic measure is non-invasive and can potentially be combined with existing, e.g., pharmacological approaches.

The complexity in analyzing the role of reactive astrocytes is reflected by the phenotypic complexity of astrocyte reactivity. Reactive astrocytes do not feature a single phenotype, which defines them as reactive, but rather multiple changes on several levels occur during the shift from the naive to the reactive state. Reactive astrocytes feature multiple phenotypes including morphological alterations, hypertrophy, increased cell migration, gene expression changes including upregulation of the intermediate filament proteins GFAP, vimentin and nestin besides others, hyperproliferation and enhanced cellular maintenance [58]. Astrocytes respond very quickly to nervous tissue injuries or other neurological abnormalities by becoming reactive within few hours after the dystrophic event [15,58]. At the lesion site also the various functions of normal, naive astrocytes are inhibited upon the shift towards a reactive cell state. Highly complex and tightly regulated functions like neuronal cell maintenance, regulation of endothelial cells at blood vessels, fine-tuning of synaptic plasticity, etc., are disturbed [2,4,5].

In the last decades, a multitude of strategies have been evaluated in order to promote functional regeneration and recovery, ranging from invasive measures over pharmacological interventions to cell intrinsic reprogramming [15,58,59,60,61]. Among the prevalent targets for pharmacological interventions are key structures in cytoskeletal stability and remodeling [62,63,64,65,66,67,68].

In the current study, we focused on increased gravitational loading induced by hypergravity as a novel non-invasive approach to investigate potential target pathways in astrocyte reactivity. Increased gravitational loads in the physiological range of 2*g* bear the potential to alter mechanosensitive structures, such as cytoskeletal components, without induction of negative effects, such as hyperproliferation or increased cell death. Most importantly, we showed that moderate hypergravity inhibited key features of astrocyte reactivity. Furthermore, in future studies the affected pathways and underlying mechanism could lead to the identification of new potential targets for neuronal regeneration to be stimulated, for instance, by pharmacological approaches. Upon identification of target transcripts, proteins, or metabolites, a combinatory approach of employing altered gravity as well as pharmacological interventions could majorly influence reactive astrogliosis first in vitro. Furthermore, studies employing 3D cultures, organoids, or organotypic cultures ex vivo should be performed before ultimately testing in vivo.

Our results suggest a role of gravitational loads on astrocyte behavior and reactivity phenotypes. Hypergravity at 2*g* did not have an extensive impact on cell morphology, but the cells were able to grow largely similarly to 1*g* control conditions (Figure 1). The MuSIC incubator-centrifuge was validated as a highly precise and reliable tool to expose cells to a variety of gravitational loading paradigms (Appendix A). Proliferation rates, cellular maintenance, and survival were undisturbed by increased gravity loads in the range of 2*g* (Figure 9). However, several mechanisms of cell movement were altered, as measured by markedly reduced cell spreading rates and cell migration speeds (Figure 2, Figure 3 and Figure 4). Importantly, acute as well as long-lasting effects could be observed on several levels. Here, astrocyte cell spreading was reduced during acute exposure to hypergravity (minutes to hours) but the effect lasted also over prolonged periods of times (up to 5 days) (Figure 2). Similarly, migration speed changes could be observed over the course of 5 days (Figure 3), but especially in live-cell imaging during initial phases (minutes to 1–2 h) of exposure to hypergravity, a pronounced reduction of spreading and migration rates was measured (Figure 4). Interestingly, the cells showed a recurrent adaptation time window (“lag phase”) of approximately 1–2 h before the respective effects were mostly apparent and stabilized. Most obvious adaptation was measured in live-cell wound-healing assays to determine migration speeds (Figure 4). The reproducible adaptation phase of astrocytes from 1*g* normal gravity migration speed to reduced speeds during hypergravity was approximately 1 h, while the re-adaptation lasted for 2 h. These values could only be obtained by introducing the novel Hyperscope platform for fast and vibration-free live-cell imaging during the exposure to hypergravity on the DLR human centrifuge (Appendix A). Decelerating migration during hypergravity is most likely due to cytoskeletal remodeling, simultaneously stabilizing microtubules and destabilizing actin filaments. The adaptation time and especially the longer re-adaptation might suggest reduced cytoskeletal rearrangement activities. In addition, exposure to altered gravity is known to induce changes in the cytoskeleton of several cell types investigated in previous studies [41,69,70].

To verify this assumption, we used the transgenic LifeAct-GFP expressing mouse line to visualize real-time F-actin dynamics during the transition from normal to increased gravity loads (Figure 6). The results from live-cell imaging during hypergravity exposure employing the LifeAct-GFP expressing cells indicated disturbed actin rearrangements, especially at regions of high turnover rates, such as migrational zones. The retraction of lamellipodia was nearly instantaneous, lacking an obvious lag phase contrary to observed lag phases for the spreading or migration rates (compare Figure 2 and Figure 4). On the contrary, stress fibers appeared mostly unaffected. Focal adhesions were formed and FAK expression levels did not change (compare Figure 8), but they re-localized toward the soma (Figure 6). Thus, highly dynamic structures and stabilization of newly formed actin filaments might be disturbed, while already stabilized structures remain intact.

The finely regulated networks of actin filaments and microtubules were thus examined in detail. Super-resolution microscopy poses a unique possibility to visualize structures beyond the refraction barrier with highly specific fluorescent multichannel labeling. We visualized actin filament and microtubule networks, using STED super-resolution microscopy and identified changes in cytoskeletal network integrity (Figure 7). Less centralized and perinuclear actin stress fibers, but denser and extensively protruding microtubules could be observed. Subcellular compartments of highly dynamic cytoskeletal rearrangements, such as migrational zones, were mostly affected by hypergravity. Microtubules appeared to be stabilized and elongated, while actin filaments showed potentially reduced stability in zones of highly dynamic rearrangements. The microtubule elongation and F-actin retraction was gravity load-dependent, as the phenotype became even more pronounced under 10*g* compared to 2*g* conditions.

Cytoskeletal architecture thus changed due to increased gravitational load, which was reflected in changes in cell morphology as well (Figure 5). Astrocytes under 2*g* hypergravity depicted less polarized and more regular cell shapes. This opposes the increased polarization and hypertrophy observed in reactive astrocytes. Reactive and hypertrophic cells often feature multiple and lobed nuclei, which was also not observed under hypergravity (Figure 10). Conclusively, morphological alterations under hypergravity support the findings that astrocyte reactivity is attenuated by 2*g* hypergravity (Figure 12).

Attenuation of astrocyte reactivity was measured as well on the molecular level by assessing protein expression levels of GFAP and vimentin both by western blot and immunofluorescence signal intensities (Figure 8 and Figure 11). The levels of GFAP, the key marker for astrocyte reactivity, were not significantly changed but rather even slightly decreased following hypergravity. The levels of other marker proteins involved in reactive astrogliosis, i.e., vimentin, nestin, and LZK, similarly showed no significant change in consequence to hypergravity, which was the case for vimentin (Figure 8 and Figure 11), or a decrease in fluorescence intensity, which was apparent for nestin and LZK (Figure 11). Therefore, the decrease of GFAP, nestin, and LZK expression following hypergravity reinforced the conclusion that astrocyte reactivity is attenuated by hypergravity in the physiologically relevant range of 2*g*.

In the current study, we aimed to investigate the potential of increased gravitational load as a measure for mechanical stimulation of cytoskeletal structures on the complex phenotype of astrocyte reactivity. Hypergravity in the range of 2*g* exerted not only no harmful effects on the cells but attenuated astrocyte reactivity on several levels (Figure 12). Importantly, astrocytes under hypergravity were able to become reactive, but the induction of the shift towards the reactive state was diminished as compared to the known progression of astrocyte reactivity induction in vivo. In order to study astrocyte reactivity, primary cells are necessary, as tumor-derived cell lines will not undergo the phenotypic shift toward the reactive state. This is of high importance for future studies employing ex vivo or in vivo models, particularly brain organoids, since an attenuation of reactive astrogliosis will greatly enhance neuronal regeneration progression, while a complete ablation of astrocyte reactivity would induce detrimental effects and widespread of the underlying injury or inflammation. Hypergravity exposure is thus a potential new non-invasive approach that can be easily combined with existing therapeutic approaches to manipulate astrocyte reactive states. Diminishing astrocyte reactivity levels in a controlled fashion by targeting the key pathways and molecules that are affected by hypergravity and/or by novel pharmacological agents could diminish the adverse effects of reactive astrogliosis during the glial scarring process and thus fundamentally enhance neuronal regeneration. This potential new therapy could aid a multitude of patients, which were subjected to neuronal injuries, trauma, infections, or suffering from neurodegenerative diseases.

## 5. Conclusions

The attenuation of key features of astrocyte reactivity due to exposure to hypergravity indicates a novel role of gravitational loading conditions on neuronal cell behavior. Especially changes in the dynamic remodeling of cytoskeletal structures were identified upon exposure to hypergravity. Deciphering the underlying mechanisms mediating these cytoskeletal changes as well as the inhibitory role hypergravity plays on astrocyte reactivity features might lead to the identification of novel targets and pathways associated with these processes. Thus, hypergravity could serve as a tool for future studies in different model systems, such as 3D cultures, organoids, or organotypic ex vivo cultures before ultimately investigating the effects in vivo. Here, the development of pharmacological agents that mimic the influence of increased gravitational loads could have high potentials for future therapeutic approaches to inhibit astrocyte reactivity. Attenuating, but not preventing, astrocyte reactivity is the crucial prerequisite for the constraint of severe astrogliosis and further the formation of the glial scar. Ultimately, attenuating astrogliosis in injured neuronal tissues would enhance the regeneration of neuronal projections and potentially restore function in areas of previously disturbed neuronal transmission. A multitude of neuronal disorders require the fight against insufficient neuronal regeneration following an injury, trauma, infection, stroke, or cancerous growth. The prospect of possible new therapies based on alterations induced by a moderate stimulus, such as increased gravity, promises enormous potential to treat patients worldwide.

## Figures and Tables

**Figure 1 biomedicines-10-01966-f001:**
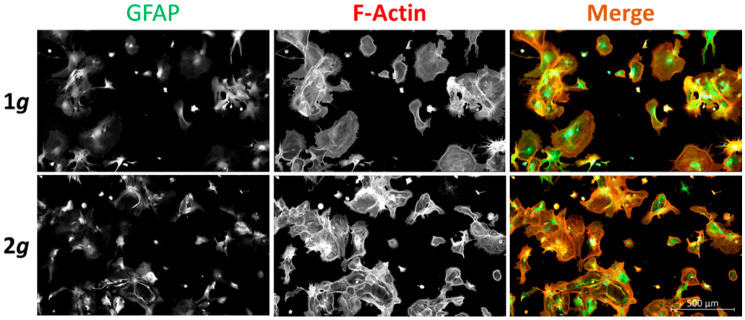
Immunofluorescence staining of primary murine astrocytes cultured at 1*g* gravity or 2*g* hypergravity. Primary murine cortical astrocyte cultures, exposed to 1*g* (top row) and 2*g* hypergravity (bottom row) for 24 h, were immuno-labeled for the intermediate filament protein glial fibrillary acidic protein (GFAP, green); F-actin was visualized using fluorescently labeled phalloidin (red). The nucleus was stained with DAPI (blue). Bar: 500 µm.

**Figure 2 biomedicines-10-01966-f002:**
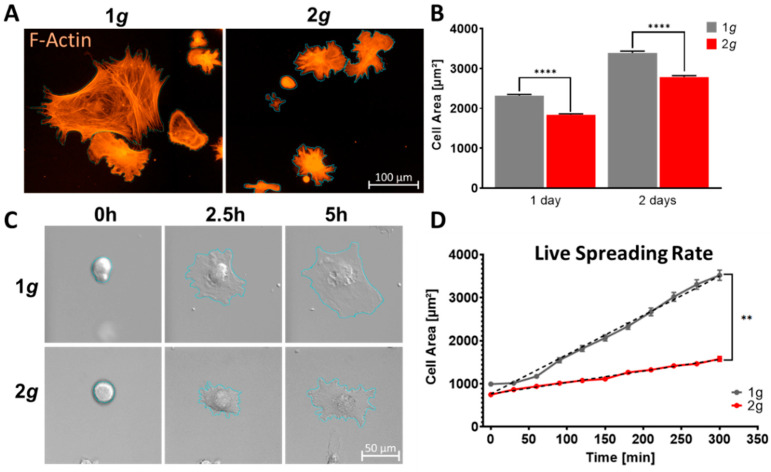
Increased gravitational load inhibits astrocytic cell spreading. (**A**) Astrocytes of the 1*g* control sample and after one day of 2*g* hypergravity exposure stained with fluorescently labeled Phalloidin (F-actin, red). The cell perimeter as recognized by semi-automatic Zeiss Zen image analysis is marked (blue). (**B**) Average cell area in µm^2^ of astrocytes at 1*g* (grey) versus 2*g* hypergravity (red) exposure (1d: *p* < 0.0001; 2d: *p* < 0.0001). At normal gravity, astrocytes occupied large areas of 2320 µm^2^ ± 29 µm^2^ compared to 1839 µm^2^ ± 22 µm^2^ at 2*g* hypergravity after 24 h. The cells enlarged further in the course of 48 h with 3392 µm^2^ ± 43 µm^2^ at 1*g* and 2785 µm^2^ ± 34 µm^2^ at 2*g* hypergravity. The samples were compared via *t*-test. The sample size n for 1 day is: 1*g* = 2320; 2*g* = 2218; for 2 days: 1*g* = 2572; 2*g* = 2491 cells from 3 individual astrocyte cultures derived from 3 gravid mice. (**C**) DIC live microscopy images of freshly seeded astrocytes at 0 h, 2.5 h, and 5 h after seeding. Top row shows the 1*g* control cells, bottom row shows cells exposed to 2*g* hypergravity on the Hyperscope live-cell imaging platform on the DLR human centrifuge. Cells which were adhering and spreading in hypergravity conditions exhibited a smaller cell area compared to control cells. (**D**) Initial spreading rate of primary astrocytes. Shown is the average cell area in µm^2^ of astrocytes in 2*g* hypergravity (red) versus 1*g* of the control (grey). A linear regression (dashed line) was inserted and slopes of the lines were calculated with 9.185 at 1*g* (y = 9.185x + 764.5) and 2.664 (y = 2.664x + 763.9) at 2*g*. If the overall slopes were identical, there was less than a 0.01% probability (*p* < 0.0001) of random data points with these different slopes. Cells exposed to hypergravity exhibit a slower increase in their average cell area. A Mann–Whitney U test showed a significant difference of the two samples (*p* = 0.0083). Values are shown as SEM and significance was indicated as follows: *p* > 0.05 as n.s., *p* < 0.05 as *, *p* < 0.01 as **, *p* < 0.001 as *** and *p* < 0.0001 as ****. The sample size n is 1*g* = 136; 2*g* = 131 cells from 3 individual astrocyte cultures derived from 3 gravid mice.

**Figure 3 biomedicines-10-01966-f003:**
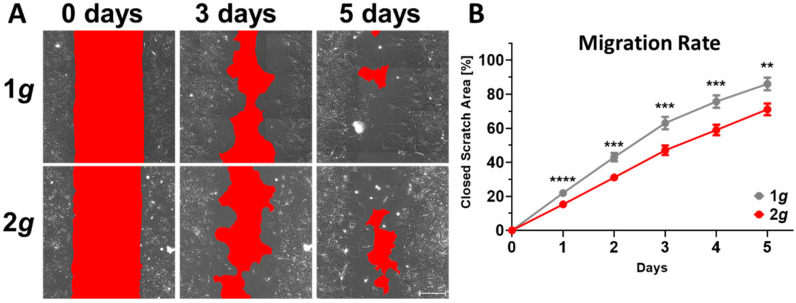
Reduced migration of astrocytes exposed to 2*g* hypergravity is a long-term effect. (**A**) Representative phase contrast microscopy images of an astrocyte wound-healing assay. Top row shows the 1*g* control sample after 0, 3, and 5 days of incubation, with the cell free area marked in red. Bottom row shows the same time points of an area that was incubated under 2*g* hypergravity in the MuSIC incubator centrifuge. Bar: 200 µm. (**B**) Line graph showing the average closed cell-free area of 2*g* hypergravity exposed astrocytes (red) and a 1*g* control (grey) over a period of 5 days. For each time point, a *t*-test was performed (day 1: *p* < 0.001; day 2: *p* = 0.002; day 3: *p* = 0.008; day 4: *p* = 0.008; day 5: *p* = 0.0049). Values are shown as SEM and significance was indicated as follows: *p* > 0.05 as ns, *p* < 0.05 as *, *p* < 0.01 as **, *p* < 0.001 as *** and *p* < 0.0001 as ****. The sample size n is 1*g* = 28; 2*g* = 32 separate wound-healing areas each with a dimension of 4 mm × 0.5 mm from 3 individual astrocyte cultures derived from 3 gravid mice.

**Figure 4 biomedicines-10-01966-f004:**
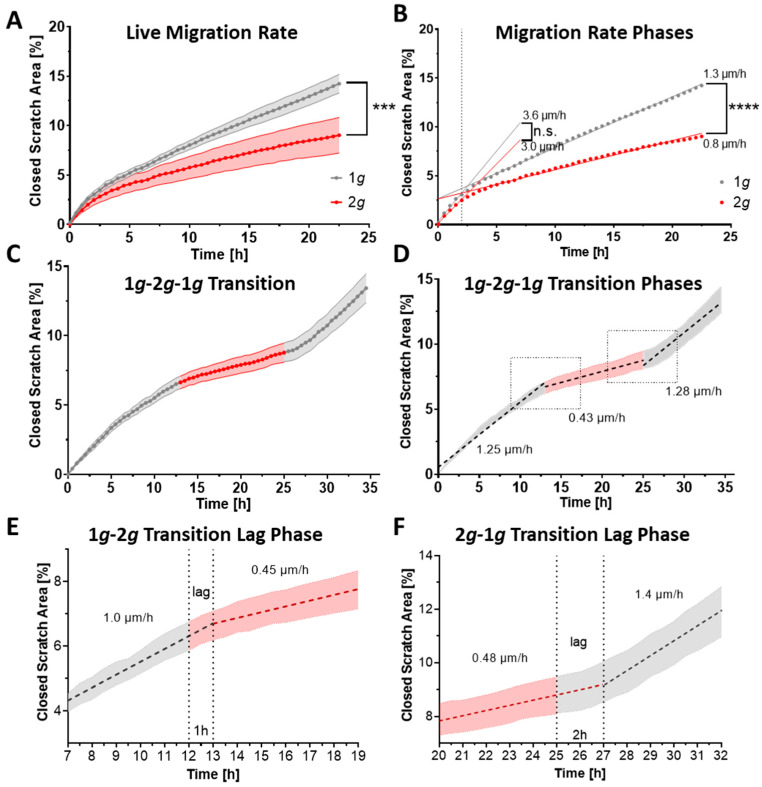
Acute effects and adaptation of the initial phase of astrocyte migration velocities with respect to gravity conditions. (**A**) Analysis of live-cell imaging of astrocytes on the Hyperscope microscope platform on the DLR human centrifuge revealed a line graph showing the average closed cell-free area of astrocytes exposed to 2*g* hypergravity (red) and 1*g* control cells (grey) over the initial 22 h of exposure. The two curves were compared with a Mann–Whitney U test (*p* = 0.005). (**B**) Linear regression for the time points 0–2.5 h and 2.5–22 h with the respective migration velocities of each regression line marked on the graph. The velocities of the 2*g* and 1*g* samples between 0 h and 2.5 h were not significantly different, in contrast to the velocities of the 2*g* and 1*g* samples from 2.5 h to 22 h (*p* < 0.0001). The sample size n was 1*g* = 6; 2*g* = 6. Separate wound-healing areas each with a dimension of 4 mm × 0.5 mm from 3 individual astrocyte cultures derived from 3 gravid mice. (**C**) Intermittent wound-healing assay with live-cell imaging during hypergravity exposure on the Hyperscope platform on the DLR large human centrifuge. Line graph showing the increase in average closed scratch area over the time course of the experiment. The scratch was imaged under 1*g* normal gravity for 12 h (grey) followed by 12 h at 2*g* hypergravity (red). The last 12 h the cells were allowed to re-adapt to 1*g* normal gravity (grey). (**D**) Linear regression lines fitted to each segment of the experiment (dashed lines) with the corresponding slope values (i.e., migration velocity) of 1*g*: 1.25 µm/h, 2*g*: 0.43 µm/h, and further 1*g*: 1.28 µm/h below. (**E**) Zoomed in line graph showing a twelve-hour period between 7 and 19 h of the wound-healing assay on the Hyperscope. Depicted is the average closed scratch area overlaid with the linear regression lines and the respective velocities noted above both lines. A one-hour lag phase was identified, which showed a steady progression of migration velocity over 1 h before the cells adapted to the hypergravity conditions with reduced migration speeds as indicated above the lines. (**F**) Similar line graph showing the period of re-adaptation from 2*g* (red) to 1*g* (grey) with the linear regression lines and their velocities noted above the lines. A lag phase of 2 h needed for re-adaptation from 2*g* hypergravity to 1*g* normal gravity was observed as indicated by the different migration velocities. The sample size n is 1*g* = 8; 2*g* = 8 separate wound-healing areas each with a dimension of 4 mm × 0.5 mm from 2 individual astrocyte cultures derived from 2 gravid mice. Values are shown as SEM and significance was indicated as follows: *p* > 0.05 as ns, *p* < 0.05 as *, *p* < 0.01 as **, *p* < 0.001 as *** and *p* < 0.0001 as ****. Tiled images of the cell-free areas were acquired with a 20× objective (NA 0.4), with increments of 30 min over the course of 36 h.

**Figure 5 biomedicines-10-01966-f005:**
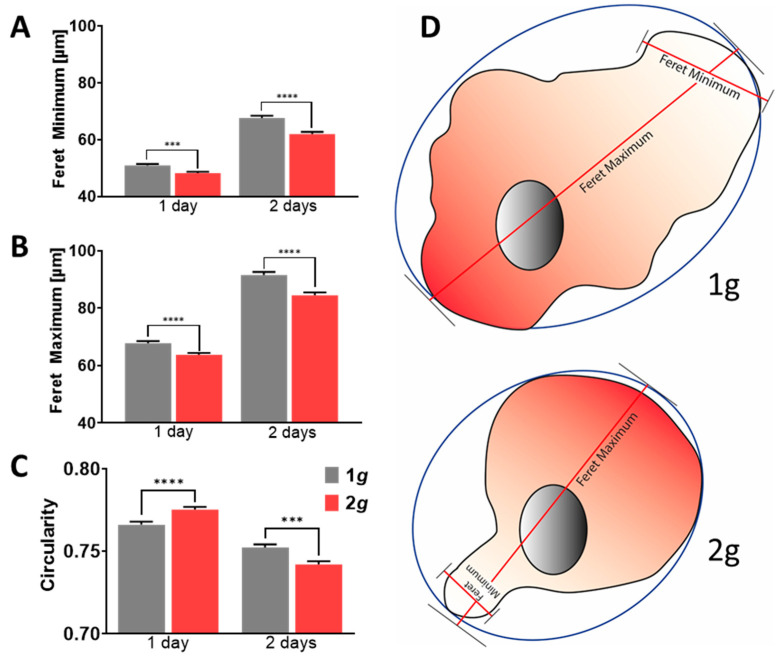
Reorganization of astrocyte cell morphology under 2*g* hypergravity. Morphological assessment of astrocytes in culture by image analysis of individual cells outlined by fluorescently labeled F-actin (compare Figure 2A for representative images of cells in culture). (**A**) Feret minimum and maximum of astrocytes after 1 and 2 days of hypergravity (2*g*) exposure. The mean Feret minimum values are given for 1*g* control cells after 1 day (grey, 50.89 µm ± 0.54 µm) and of 2*g* hypergravity exposed cells (red, 48.22 µm ± 0.47 µm, *p* = 0.0002). Similarly, after 2 days for the 1*g* controls (grey, 67.66 µm ± 0.78 µm) and the 2*g* samples (red, 62.00 µm ± 0.74 µm, *p* < 0.0001). (**B**) Mean Feret maximum after 1 day of 1*g* control (grey, 67.77 µm ± 0.76 µm) and 2*g* hypergravity sample (red, 63.73 µm ± 0.64 µm, *p* < 0.0001), and after 2 days 1*g* control (grey, 91.49 µm ± 1.06 µm) and 2*g* exposure (red, 84.40 µm ± 1.05 µm, *p* < 0.0001). The sample size n is 1 day: 1*g* = 2320; 2*g* = 2218; 2 days: 1*g* = 2572; 2*g* = 2491 cells from 3 individual astrocyte cultures derived from 3 gravid mice. (**C**) Mean circularity of astrocytes was calculated by the formula circularity = $\sqrt{\frac{4\times Area}{\pi \times Feret\;Max2}}$ in the 1*g* control group (grey, 0.77 ± 0.01) and cells exposed to 2*g* hypergravity for 1 day (red, 0.78 ± 0.01, *p* < 0.0001) and 2 days in 1*g* (0.75 ± 0.01) and 2*g* hypergravity (0.74 ± 0.01, *p* = 0.0001). The sample size n is: 1 day: 1*g* = 2320; 2*g* = 2218; 2 days: 1*g* = 2572; 2*g* = 2491 cells from 3 individual astrocyte cultures derived from 3 gravid mice. For every time point, the two conditions were compared via *t*-test. Values are shown as SEM and significance was indicated as follows: *p* > 0.05 as ns, *p* < 0.05 as *, *p* < 0.01 as **, *p* < 0.001 as *** and *p* < 0.0001 as ****. (**D**) The schemes represent a graphical demonstration of the morphological changes that the astrocytes undergo from 1*g* normal Earth gravity to 2*g* hypergravity.

**Figure 6 biomedicines-10-01966-f006:**
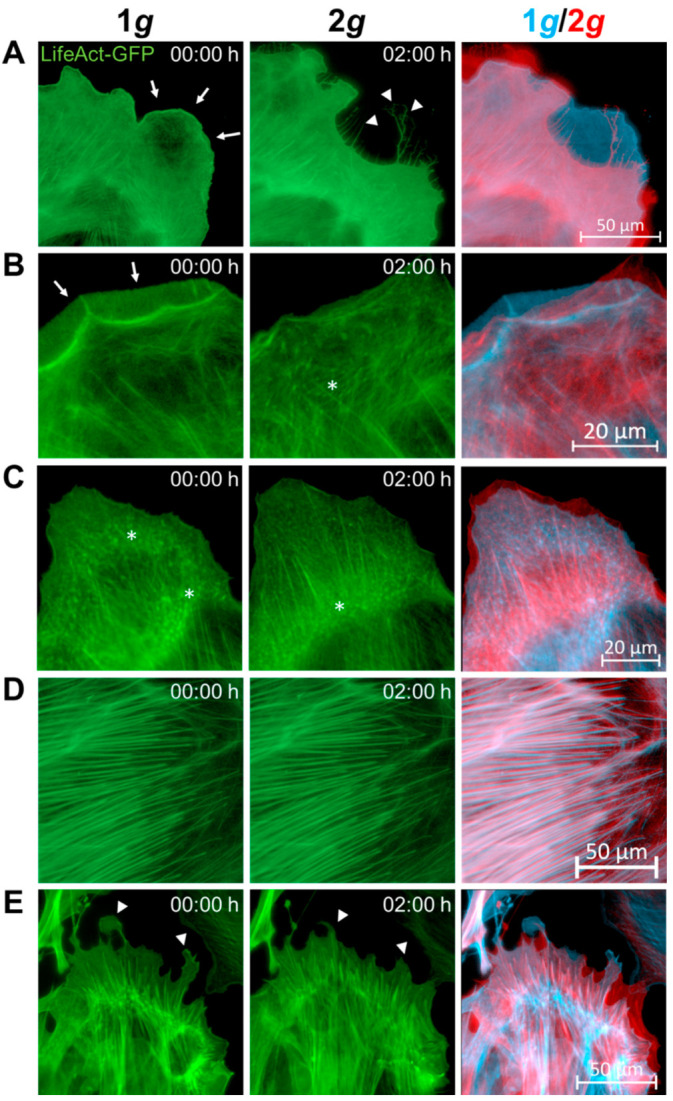
Highly dynamic F-actin rearrangements are altered due to hypergravity. Primary murine transgenic LifeAct-GFP expressing astrocytes were subjected to 2*g* hypergravity on the Hyperscope live-cell imaging platform on the DLR human centrifuge. (**A**) Astrocyte lamellipodia (arrows) retracted with filopodia (arrowheads) remaining. (**B**) Lamellipodial retraction and membrane ruffles (arrows) could be observed. (**C**) Focal adhesions (asterisks) were still able to form under the influence of hypergravity, but with a change in location towards the cell center. (**D**) Stress fibers remained intact and largely unchanged. (**E**) Even though stress fibers remained intact, retraction of larger protrusions (arrowheads) lead to rearrangements also of more stable structures. The cells were exposed to 1*g* or 2*g* hypergravity on the same setup on the Hyperscope for 3 h with 2.5 min increments between images. The left and middle images show F-actin structures of an astrocyte at 1*g* or 2*g* conditions, respectively. The right image shows an overlay of the 1*g* (blue) and 2*g* (red) image in different colors to indicate the structural changes over time as indicated by time stamps in min. Scale bars for every row are indicated in the right panels.

**Figure 7 biomedicines-10-01966-f007:**
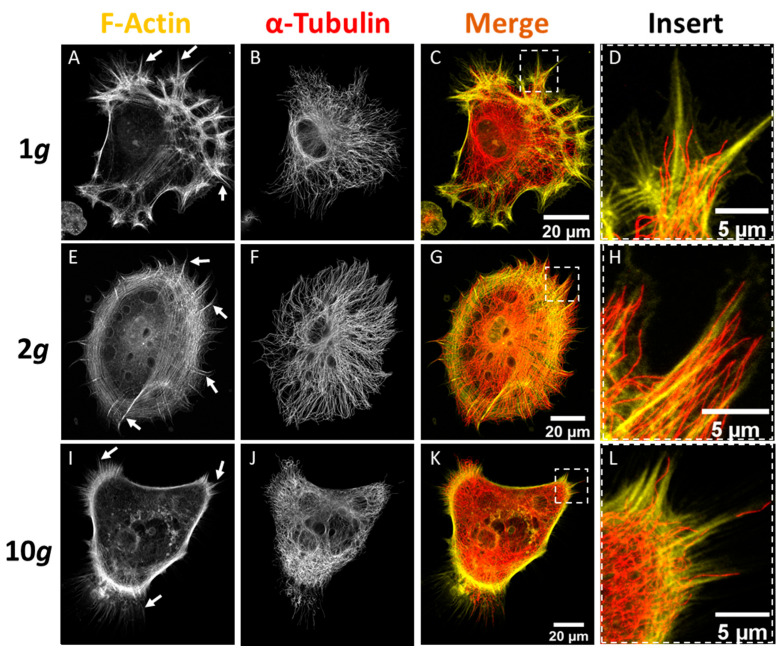
Gravity-dependent cytoskeletal rearrangements revealed by super-resolution microscopy. Super-resolution (STED) microscopy images of a representative primary murine astrocyte cultured under 1*g*, 2*g*, and 10*g* hypergravity conditions and immunostained for F-actin using Phalloidin conjugated to an Atto542 dye and for α-tubulin using a specific antibody and a secondary antibody conjugated to an Abberior STAR RED dye. Astrocytes were exposed to normal 1*g* gravity conditions (**A**–**D**), 2*g* hypergravity (**E**–**H**), and 10*g* hypergravity (**I**–**L**) in the DLR MuSIC incubator-centrifuge. The arrows mark several lamellipodia with additional filopodia extruded from the cell membrane at various points. Shown are single channels of F-actin (yellow) and α-tubulin (red) and a merged image. Marked with a box (insert) is a magnified protrusive element. The images are maximum intensity projections of 5 optical sections acquired by a z-stack of 200 nm step size.

**Figure 8 biomedicines-10-01966-f008:**
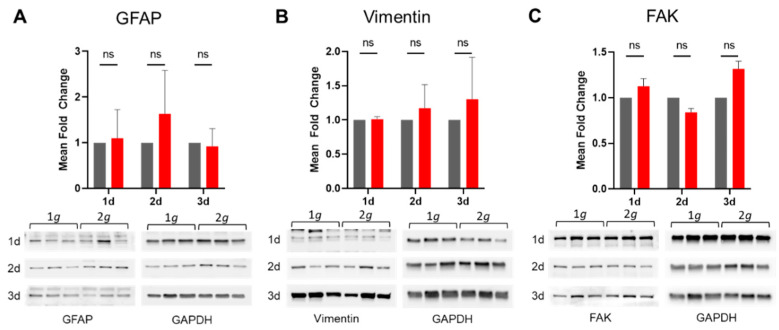
Intermediate filament and focal adhesion protein expression remain stable under the influence of 2*g* hypergravity. Western blot analysis of glial fibrillary acidic protein (GFAP, 55 kDa), vimentin (54 kDa) and focal adhesion kinase (FAK, 125 kDa). Mean fold changes of the three individual hypergravity samples are shown as compared to the matching 1*g* controls. Individual blots were obtained for each day of exposure and for each protein of interest matched with the corresponding GAPDH (37 kDa) signal detection on the same membrane. All signals were normalized to the GAPDH signal intensity as a housekeeping gene. (**A**) GFAP expression remained mostly stable under 2*g* hypergravity loads (1d *p* = 0.9437; 2d *p* = 0.6628; 3d *p* = 0.9585). (**B**) The expression levels of vimentin were slightly elevated over the course of 3 days (1d *p* = 0.9903; 2d *p* = 0.8397; 3d *p* = 0.7216). (**C**) FAK expression was stable under 2*g* increased gravitational load (1d *p* = 0.4248; 2d *p* = 0.3145; 3d *p* = 0.0741). The lysates were obtained from three individual cultures derived from three mice. The values are shown as SEM, and they were compared via *t*-test. Significance was indicated as follows: *p* > 0.05 as ns, *p* < 0.05 as *, *p* < 0.01 as **, *p* < 0.001 as *** and *p* < 0.0001 as ****.

**Figure 9 biomedicines-10-01966-f009:**
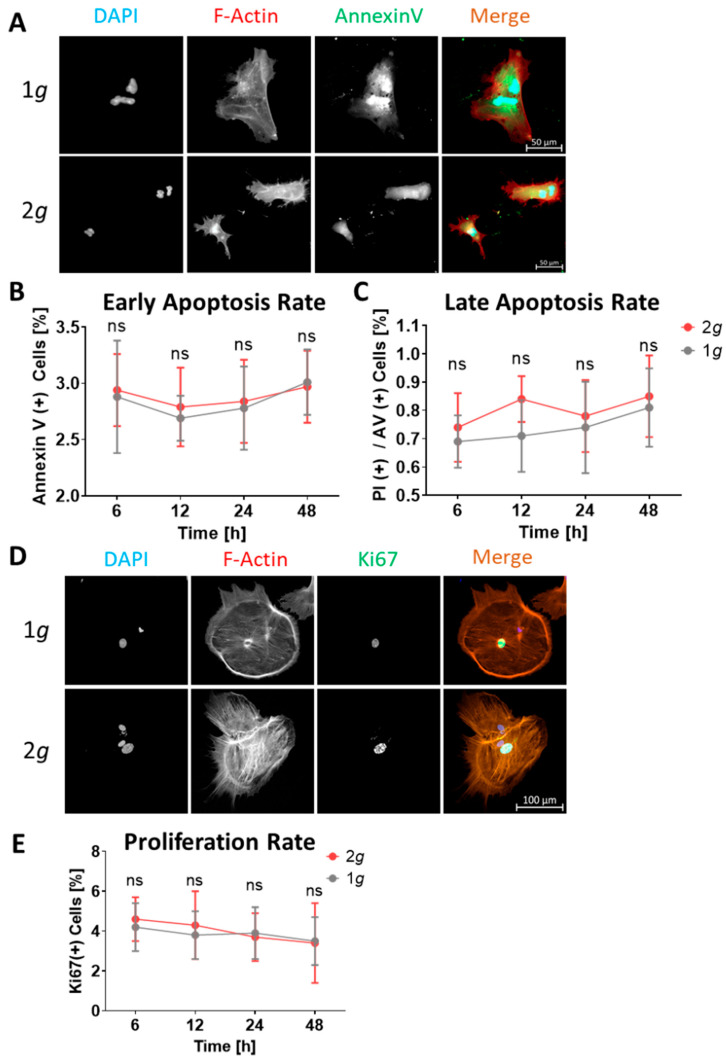
Astrocyte proliferation and apoptosis rates are not influenced by 2*g* hypergravity. Assessment of early and late apoptosis (i.e., necrosis) as well as proliferation of primary murine astrocytes over 2 days of 2*g* hypergravity exposure on the MuSIC incubator-centrifuge. (**A**) Representative immunofluorescence image of astrocytes stained with DAPI (blue), Annexin V-Atto488 (green) and Phalloidin-Atto542 labeling actin filaments (red) for 1*g* control and 2*g* hypergravity conditions. Propidium iodide was used to label necrotic cell nuclei (data not shown) (**B**) Astrocytes were exposed to 2*g* hypergravity and the number of apoptotic Annexin V-positive cells was counted and compared to 1*g* controls after 6 h (1*g*: 2.88% ± 0.29%; 2*g*: 2.94% ± 0.18%; *p* = 0.8695), 12 h (1*g*: 2.69% ± 0.20%; 2*g*: 2.79% ± 0.35%; *p* = 0.6896), 24 h (1*g*: 2.78% +/− 0.21%; 2*g*: 2.84% +/− 0.21%; *p* = 0.8523), and 48 h (1*g*: 3.01% ± 0.29%; 2*g*: 2.97% ± 0.32%; *p* = 0.8803) of exposure. (**C**) Late apoptotic (necrotic) cells were counted as Annexin V- and PI- double positive. Necrotic cells could be observed in small numbers for 6 h (1*g*: 0.69% ± 0.16%; *2g*: 0.74% ± 0.21%; *p* = 0.7593), 12h (1*g*: 0.74% ± 0.12%; *2g*: 0.84% ± 0.08%; *p* = 0.4366), 24 h (1*g*: 0.74% ± 0.16%; *2g*: 0.78% ± 0.13%; *p* = 0.8552), and 48 h (1*g*: 0.81% ± 0.24%; 2*g*: 0.85% ± 0.25%; *p* = 0.8513). (**D**) Representative immunofluorescence images of astrocytes from 1*g* controls or 2*g* hypergravity samples stained with DAPI (blue), Phalloidin-Atto542 labeling actin filaments (red), and an anti-Ki67 antibody conjugated to an Atto488 dye (green). (**E**) The number of Ki67-positive nuclei of astrocytes were counted after exposure to 2*g* vs. 1*g* after 6 h (1*g*: 4.20% ± 1.20%; 2*g*: 4.60% ± 1.10%; *p* = 0.6923), 12 h (1*g*: 3.8% ± 0.69%; 2*g*: 4.3% ± 0.98%; *p* = 0.6986), 24 h (1*g*: 3.9% ± 0.75%; 2*g*: 3.7% ± 0.69%; *p* = 8543), and 48 h (1*g*: 3.50% ± 1.20%; 2*g*: 3.40% ± 2.00%; *p* = 0.9444). Values are shown as SEM and significance was indicated as follows: *p* > 0.05 as ns, *p* < 0.05 as *, *p* < 0.01 as **, *p* < 0.001 as *** and *p* < 0.0001 as ****. For every time point the two conditions were compared via *t*-test. The sample size n is 10,549 cells (apoptosis & necrosis) and 10,977 cells (proliferation), each from three individual astrocyte cultures derived from three gravid mice.

**Figure 10 biomedicines-10-01966-f010:**
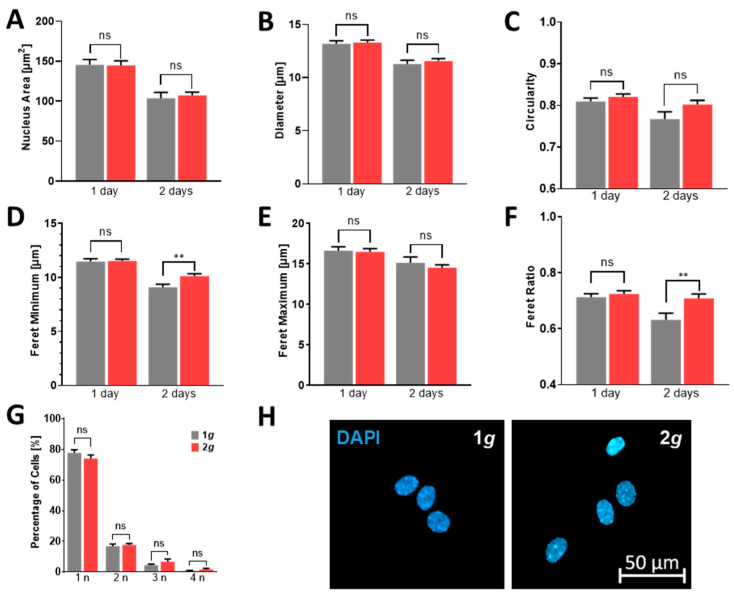
Astrocytes under hypergravity feature normal nuclear morphology and are not multinucleated. Nucleus morphology and count in astrocytes exposed to 2*g* hypergravity. (**A**) Mean area of nuclei after 1 day in the 1*g* control group (grey, 145.5 µm^2^ ± 6.7 µm^2^) and under 2*g* hypergravity (red, 144.8 µm^2^ ± 5.8 µm^2^; *p* = 0. 9372), and after 2 days at 1*g* (103.8 µm^2^ ± 7.4 µm^2^) and 2*g* (107.3 µm^2^ ± 4.1 µm^2^; *p* = 0.6533). (**B**) Mean diameter of nuclei after 1 day in the 1*g* control group (13.2 µm ± 0.3 µm) and under 2*g* hypergravity (13.3 µm ± 0.2 µm), and after 2 days at 1*g* (11.3 µm ± 0.4 µm) and 2*g* (11.6 µm ± 0.2 µm). (**C**) Mean circularity of nuclei after 1 day in the 1*g* control group (0.81 ± 0.01) and under 2*g* hypergravity (0.83 ± 0.01; *p*= 0.7866; *p* = 0.2907), and after 2 days at 1*g* (0.76 ± 0.01) and 2*g* (0.8 ± 0.01; *p* = 0.0633). (**D**) Mean Feret minimum of nuclei after 1 day in the 1*g* control group (11,5 µm ± 0.3 µm) and under 2*g* hypergravity (11.5 µm ± 0.2 µm; *p* = 0.8834), and after 2 days at 1*g* (9.1 µm ± 0.3 µm) and 2*g* (10.1 µm ± 0.2 µm; *p* = 0.0044). (**E**) Mean Feret maximum of nuclei after 1 day in the 1*g* control group (16.6 µm ± 0.5 µm) and under 2g hypergravity (16.5 µm ± 0.4 µm; *p* = 0.7895), and after 2 days at 1*g* (15.1 µm ± 0.7 µm) and 2*g* (14.5 µm ± 0.3 µm; *p* = 0.4048). (**F**) Mean Feret ratio of nuclei after 1 day in the 1*g* control group (0.71 ± 0.01) and under 2*g* hypergravity (0.73 ± 0.01; *p* = 0.4951), and after 2 days at 1*g* (0.63 ± 0.02) and 2*g* (0.71 ± 0.01, *p* = 0.0063). The sample size n is 1 day: 1*g* = 140; 2*g* = 131; 2 days: 1*g* = 39; 2*g* = 61 cells from one individual astrocyte culture derived from one gravid mouse. (**G**) Average percentage of nuclei per cell of astrocytes exposed to 1*g* control conditions (n = 844) and 2*g* hypergravity (n = 680) for up to 2 days. One nucleus was observed in 77.8% ± 2.0% of cells exposed to 1*g* and in 74.1% ± 2.3% of cells exposed to 2*g* (*p* = 0.2364). Two nuclei were observed in 16.9% ± 1.4% of 1*g* control and in 17.7% ± 0.9% of 2*g* hypergravity exposed cells (*p* = 0.6417). Three nuclei were observed in 4.5% ± 0.6% of 1*g* control and in 6.7% ± 1.7% of 2*g* hypergravity exposed cells (*p* = 0.2141). Four nuclei were observed in 0.8% ± 0.2% of 1*g* control and in 1.6% ± 0.8% of 2*g* hypergravity exposed cells (*p* = 0.2664). The sample size n is 1*g* = 844; 2g = 680 cells from 2 individual astrocyte cultures derived from 2 gravid mice and compared by *t*-test. Values are shown as SEM and significance was indicated as follows: *p* > 0.05 as ns, *p* < 0.05 as *, *p* < 0.01 as **, *p* < 0.001 as *** and *p* < 0.0001 as ****. (**H**) Representative immunofluorescence image of astrocyte nuclei stained with DAPI, under normal 1*g* gravity (left) and 2*g* hypergravity (right).

**Figure 11 biomedicines-10-01966-f011:**
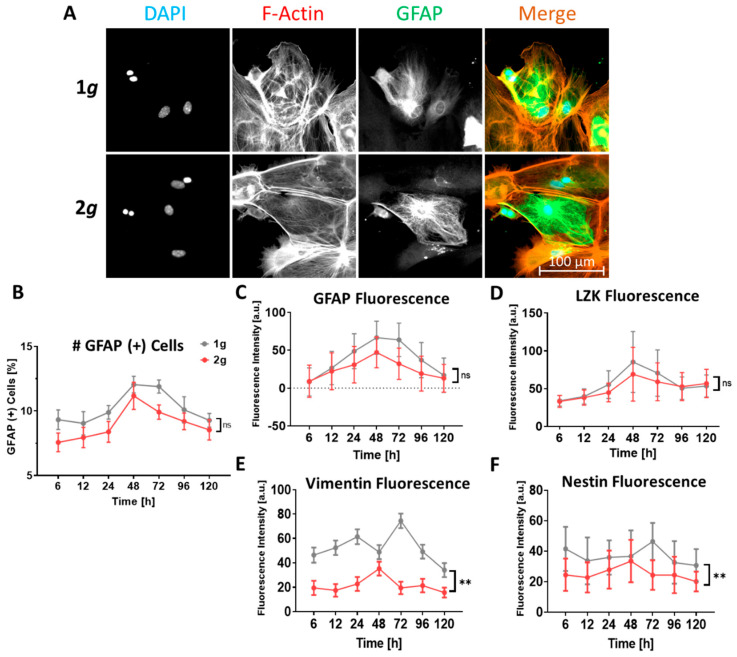
Astrocytes remain in their native homeostatic state without shifting to a reactive phenotype by physiological levels of hypergravity. Astrocytic reactivity was assessed by immunofluorescent labeling of several marker proteins of reactive astrocytes. (**A**) Representative epifluorescence microscopy image of primary murine astrocytes at 1*g* normal gravity and 2*g* hypergravity conditions. The cells were stained with DAPI (blue), an anti-GFAP antibody conjugated to an ATTO488 dye (green) and Phalloidin conjugated to an ATTO542 dye labeling F-actin filaments (red). (**B**) Cells that were expressing high levels of GFAP were counted as GFAP-positive cells. The line graph depicts the mean fraction of GFAP-positive astrocytes exposed to 1*g* control or 2*g* hypergravity conditions (*p* = 0.0728). (**C**–**F**) Fluorescence intensity of glial reactivity markers after hypergravity exposure were measured for individual cells. The cells were semi-automatically traced and their mean fluorescence intensity was measured in arbitrary units (a.u.). The cells were labeled by antibodies specific for GFAP and LZK or vimentin and nestin in separate stainings to avoid overlapping of fluorescent channels. Mean values for the 1*g* and 2*g* hypergravity samples are shown as SEM and compared by Mann–Whitney U test. *P*-values were calculated for GFAP (*p* = 0.3176), LZK (*p* = 0.7104), vimentin (*p* = 0.0012), and nestin (*p* = 0.0023) and significance was indicated as follows: *p* > 0.05 as ns, *p* < 0.05 as *, *p* < 0.01 as **, *p* < 0.001 as *** and *p* < 0.0001 as ****. The sample size n was 20,159 cells (GFAP), 19,777 cells (LZK), 20,821 cells (nestin), and 21,426 cells (vimentin) each derived from 3 individual astrocyte cultures derived from 3 gravid mice.

**Figure 12 biomedicines-10-01966-f012:**
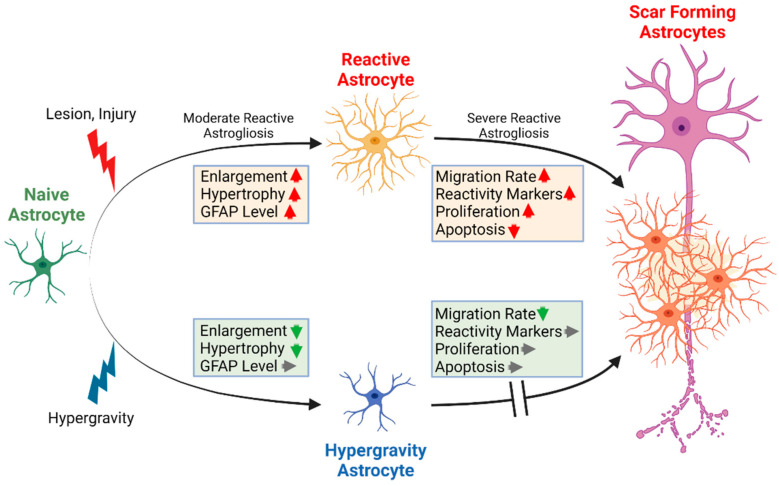
Schematic overview of the process of astrocyte reactivity induction and the influence of hypergravity.

## Data Availability

Not applicable.

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
