# Peer review of "Hypergravity Attenuates Reactivity in Primary Murine Astrocytes"

_biomedicines, 2022, doi:10.3390/biomedicines10081966_

Round 1

Reviewer 1 Report

The work entitled “Hypergravity Attenuates Reactivity in Primary Murine Astrocytes” provides evidence about the implication that moderate hypergravity inhibited the critical features of astrocyte reactivity. Reactive astrocytes due to nervous tissue injury promote neuronal regeneration and lead to astrogliosis and glial scar formation damage to the tissue and subsequently impairments in cognitive and/or motor functions. This study demonstrated that hypergravity exposure partially inhibits astrocyte reactivity by spreading deficits, impaired migration speed, decreased cell polarity actin filament and microtubule dynamics, and significant cytoskeletal expression experiments. Hence, hypergravity would be a non-invasive novel tool for future studies on identifying the key pathways and molecules for pharmacological interventions.

The article is well written and structured except for a few minor comments, as mentioned below.

Comments and Suggestions for Authors:

Page 2, Line # 66: NG2 cells- abbreviate

Page 3, Line # 141 Here, the authors mentioned the cells are exposed to 2 g or 10 hypergravity, but the exposure to 10 hypergravity was used only in one experiment on Page 17, line #646, and on Page 18, fig 7 and the line #673. What basis was 10 hypergravity exposure selected? Why not use 10 hypergravity exposure in other experiments? 

Page 18 Fig 7. The figure showed the insert but no mention of the insert in legend or text.

Page 19 Fig 8. Too many gel pictures in one figure. Received the original WB gels images in email attachments and verified. 

Author Response

Point-by-Point Response to Reviewers

Hypergravity Attenuates Reactivity in Primary Murine Astrocytes

 Yannick Lichterfeld, Laura Kalinski, Sarah Schunk, Theresa Schmakeit, Sebastian Feles, Timo Frett, Harald Herrmann, Ruth Hemmersbach and Christian Liemersdorf*

28.07.2022

 Dear editors and reviewers,

we would like to express our gratitude to your comments and suggestions that we were happy to oblige in our efforts to further improve the manuscript.

All authors had the opportunity to reply to the reviews and add to our response. We would like to submit the revised version of the manuscript in the “track-changes” mode as a MS Word file as well as this point-by-point response letter to the reviewers with our answers written below the original comments in blue.

If you have any further questions or comments, please feel free to get in contact with us and thank you once again for your efforts!

Sincerely,

Christian Liemersdorf

Reviewer 2 Report

This manuscript describes development of cell culture imaging system on large-scale centrifuge platform to assess the impact of hyper gravity. Using this platform, the authors examined impact of 2g environment on mouse primary astrocyte culture and induction of reactive astrocyte. Automated microscope system on a human-size centrifuge allowed the authors to capture multi-day experiments in which cellular behavior was captured while the cells are under continuous and stable hyper gravity environment. Cellular model used was also biologically interesting. Accurate evaluation of hyper gravity was previously very difficult. Thus, this study provides extremely important data to the field.

There were only two minor points to suggest for improvements:

1.      Quantification in Figure 5 seems to detect morphological changes in detail, and drawing of typical cell model was very helpful. It would be even better, if it is possible, to add typical cell image to this figure.

2.      Early vs late cell (early = apoptosis, later = necrosis?) argument was unclear with current description (lines 737-745). It would be helpful if the authors could re-write this section. The argument seems fine, but it was just difficult to fully appreciate what the authors were trying to explain.

Author Response

(The authors gave the same response as above.)

Reviewer 3 Report

These are results of primary importance concerning the effects of changes in gravity on the dynamics of cytoskeletal movements (in particular during the acceleration phase and during stable hypergravity). These results need to be better presented. The introduction mainly reports related works it seems important to me to report the work concerning gravity and cellular dynamics (Vassy et al., 2001 doi :10.1096/fj.00-0527fje, experiments realized in FLUMIAS, a DLR instrument). The part concerning pathology is particularly long and without real interest in the problematic .this MS reports a major work in the field of gravity impact on biology. The alterations in the cerebral functions of subjects subjected to repeated gravitational changes (astronauts, pilots, sports drivers, etc.) when they exist are not related to neurodegenerative pathologies. However, since the 1980s, numerous works have demonstrated that astrocyte plasticity was necessary for the plasticity of neural networks in neuroendocrine, sensory motor and cognitive functions, it seems to me that building the introduction around these aspects would increase the scope of the publication. Finally, introduction can also point the mechanosensitivity and mechanotransduction in astrocytes (what is known, suggested or not).

In 2.2.2.2. is it possible to indicate the duration of the experiment (indicate the duration during which the cells are no longer subjected to hypergravity; cells are outside the hypergravity device during XX ± yy min).

In 2.3.4 the macro, pluging or algorythms used in ImageJ should be indicated and detailed to give the opportunity to the readers to reproduce the same analysis to compare their data to those developed in this MS.

authors should standardize the way they report results. There are values in the text without statistical indication which are present in the legends, sometimes all the numerical indications are in legends. A more orthodox way is to write a descriptive text of the results containing the average and statistical values and a more concentrated legend. In addition, illustrations of the protocols on the figures would provide visual access to them, making the reading of the figures easier.

Figure 2B a two-way ANOVA might give more information and is more appropriated.

Figure 2D, the analysis of the slope of the straight lines and the statistics on each pair of points would make it possible to suggest when the two curves diverge. The equation of the lines could be added in the text. Put the stats in the text.

Results concerning velocities of migration could be more precised with SD or SEM (lines 433-434).

Fire 5, the Two-way analysis should be used to precise interaction between duration and gravity effects

Line 657 “microtubules innervated” can you precise or use another word as “structurating”

Figure 7, is it possible to quantify the F actin and tubulin networks ? (density of fiber) length of fibers ?

Figure 8: there are many variations in band intensities as the ponceau red was made to relate the densities of the bands to the total protein signal should be better (as required by several publication about WB quantifcation). 2way anova should be more informative to evaluate interaction between gravity level and time exposures.  But I have two comments and on conclusion on this Figure. 1/ If I have well understood the mean fold change is a ratio ratio, and for me statistics are not possible on this kind of results. 2/ I have analyzed the images of the WB membranes attached to the MS. There revealed some problems for me : It seems that membranes were not imaged with the same device (there are some images with black bands on white and some others are white bands on black background). The size of the molecular weight markers is missing for easy reading of the data. There are several bands for each well that presages either a lack of specificity of the antibodies which is very embarrassing or a too large quantity of primary or secondary antibody for the experiment (that can increase binding on other epitopes). This would require verification of bands created with secondary antibodies alone and justification of the concentration of primary antibody used in the experiments. Lower molecular weight bands may be produced by degradation of the target protein. In this case, there it would be necessary at least a quantification of these bands. For the bands with higher molecular weight, it would be necessary to specify if this can come from dimerization or protein associations insensitive to the treatment used for protein preparation. Anyway, if there is quantification from these membranes it must be done according to the most rigorous standards, i.e. a density of each band compared to the quantity of protein deposited in each well (see reference publication about WB quantification) and of course there should be no overdriven band. My conclusion this experiment is not relevant for the demonstration of the cytoarchitecture modifications induced by hypergravity, the other experiments are more convincing. Then the Figure 8 could be withdrawn.

Figure 9, could you add illustration and graph concerning PI alone ?

Figure 10, could you please add two-way anova analysis ?

Figure 11, could you detail the method to ensure that there is no signal saturation. Here a WB for these markers under these experimental conditions at 24h and 120h for example to confirm the result should be more informative (I don’t require it).

 Line 997, “astrocytes respond very quickly to nervous tissues injuries” for sure but it this case I think it is better to discuss the results regarding experiments proving that astrocytes were engaged in physiological responses especially in synaptic plasticity. These work were important (for me the first one is from Theodosis group: Montagnese et al., 1987).

Line 1039 The authors can reinforce this very important point because it is crucial for the phase of gravity changes during space flights or other human activities with acceleration phases.

 Figure 12: to arrive at this schema, a dataset on lesion astrocytes would have been relevant. Can scratch be considered a lesion? if yes indicate it in the progress of the MS. If the discussion connects figures 8 and 11 permanently why not make only one figure?

Author Response

(The authors gave the same response as above.)

Round 2

Reviewer 3 Report

I have carefully read all your justifications and I thank you for them. I think an explanation of the statistical choice might make the MS even more supporting (including your explanation from your answer). this seems important to me so that readers do not doubt the rigor of the analysis. In the same way the choice concerning the quantification of the WB should be mentioned. Thank you for all of your comments.

best regards